

# Large scale hydrological model river storage and discharge correction using satellite altimetry-based discharge product

Charlotte Marie Emery [1,*], Adrien Paris [2,3], Sylvain Biancamaria [1], Aaron Boone [4], Stéphane Calmant [1], Pierre-André Garambois [5], and Joecila Santos da Silva [6]

[1]LEGOS, Université de Toulouse, CNRS, CNES, IRD, UPS, Toulouse, France
[*]now at: JPL, Pasadena, CA, USA
[2]GET, Toulouse, France
[3]LMI OCE IRD/UNB, Campus Darcy Ribeiro, Brasilia, Brazil
[4]CNRM, Toulouse, France
[5]ICUBE-UMR 7357, Fluid Mechanics Team, INSA, Strasbourg, France
[6]CESTU, Universidade do Estado do Amazonas, Manaus, Brazil

*Correspondence to:* Charlotte Marie Emery (charlotte.emery@jpl.nasa.gov)

**Abstract.** Land Surface Models (LSM) are widely used to study the continental part of the water cycle. Yet, even though their accuracy is increasing, inherent model uncertainties can not be avoided. In the meantime, remotely-sensed observations of the continental water cycle variables such as soil moisture, lakes and rivers elevations are more frequent and accurate. Therefore, those two different types of information can be combined, using data assimilation techniques to reduce model's uncertainties on its state variables or/and on its input parameters. The objective of this study is to present a data assimilation platform that assimilates into the LSM ISBA-CTRIP a river discharge product, derived from ENVISAT nadir altimeter water elevation measurements, over the whole Amazon basin. The study presents an initial development for a localization treatment that allows to limit the impact of observations to areas close to the observation and on the same hydrological network. This assimilation platform is based on the Ensemble Kalman Filter and can either correct the river water storage or the discharge: RMSE compared to gauge discharges are globally reduced until 21% and at Óbidos, near the outlet, RMSE are reduced up to 52% compared to ENVISAT discharge. Finally, it is shown that localization improve results along main tributaries.

## 1 Introduction

The continental part of the water cycle is commonly studied, at large scale, with hydrological modelling. These models are generally issued from the coupling of a Land Surface Model (LSM) with a River Routing Model (RRM). The LSM determines the water and energy budget at the surface by spreading precipitations between the soil and the canopy. Meanwhile, the RRM transfers water mass through the basin to the outlet and gives an estimate of river discharge.

RRMs are mainly based on kinematic (Oki and Sud, 1998) or sometimes also on diffusive wave model (Yamazaki et al., 2011). The routing network is derived from a Digital Elevation Model (DEM). Several RRMs were developed during the 90s. They differ from one another by their surface parametrization, the flow velocity modelling and the inclusion or not of



groundwater and floodplains dynamics (Oki and Sud, 1998; Coe, 1998; Hagemann and Dümenil, 1998; Ducharne et al., 2003; Ngo-Duc et al., 2007; Decharme et al., 2012).

However, even if hydrological models become more and more accurate, inherent model uncertainties are unavoidable. They originate from several sources. First, the real physics is still not completely known. Therefore, to build the model's equations,
some simplification assumptions have to be considered. New uncertainties are introduced during the model numerical solving due to time and space discretization and because there is generally no analytical solution to the equations. Finally, all models use a high number of inputs (parameters and variables). Exact values of these inputs are not known and some a priori values are used. All these uncertainties impact model's outputs. The model gives therefore an approximative view of the system real state.

Observations of the system can be used to calibrate and/or validate the model and reduce its errors. These observations can be obtained from in situ or remote techniques. In situ techniques mainly focus on measuring river water elevations at a gage station. Another important variable of interest in river hydrology is the river discharge, which is sparsely measured compared to water elevation. Based on river discharges and elevations measured at the same time and at the same location, it is possible to build a rating curve that represents the elevation-discharge relationship. This rating curve is then applied to water
elevation to set continuous discharge time series. Institutions delivering in situ data provide mainly discharge. Even though in situ measures are generally quite accurate with a high time sampling (i.e. sub-daily), their main limitations is their local and spatially sparse sampling over the river network. Furthermore, nowadays, remotely-sensed data from satellite missions are more and more available and provide useful observations of rivers. The most straightforward and used instrument to measure river water elevations is the nadir altimeter.

Altimeters were initially developed to measure oceans topography with the satellite mission GEOS-3 (1975-1978) and SEASAT launched in 1978 (MacArthur, 1980). Nadir altimetry consists in estimating water surface elevation at the vertical ( or at the nadir ) of the satellite. It therefore produces punctual water elevation observations along the satellite ground track. These missions were followed by a long series of other missions: GEOSAT (1985-1990), TOPEX-Poseidon (1992-2006), JASON-1 (2001-2013), JASON-2 (2008-Now), JASON-3 (2016-Now), GFO (1998-2008), ERS-1 (1991-2000), ERS-2 (1995-
2003), ENVISAT (2002-2012), SARAL (2013-Now) and Sentinel-3 (2016-Now). It is with TOPEX-Poseidon that the use of nadir altimetry to monitor lakes (Birkett, 1995; Hwang et al., 2005; Cretaux et al., 2009), floodplains (Birkett et al., 2002) and rivers (Birkett, 1998; Kouraev et al., 2004; Silva et al., 2010) developed widely. However, the main limitations of nadir altimetry are their punctual measurements (at the location where the satellite track crosses a river stream) and their temporal sampling (from 10 to 35 days, depending on the mission). Besides, contrary to ocean surfaces, the signal over continental
surfaces is impacted by vegetation and topography surrounding the river. Therefore, the purpose of this study is to combine model outputs and altimetry-based products using Data Assimilation (DA) techniques, in order to get more precise discharge estimates within the Amazon basin.

DA aims to improve model skills to forecast/simulate the physical system evolution. To do so, DA techniques focus on either correcting the model's input parameters (Parameter Estimation) or the model's outputs (State Estimation). State Estimation
(SE) consists in using observations to directly correct the model output state. It is based on the assumption that the model (and





the observations) are known to be imperfect. So, SE aims at correcting model outputs, which errors result from all sources of uncertainties previously described.

SE has been widely used in oceanography and meteorology (Evensen and Leeuwen, 1996; Houtekamer and Mitchell, 2001; Gustaffson et al., 2012; Tanajura et al., 2015). However, DA of remotely-sensed observations to correct hydrological models states is more recent. Moreover, it is more developed for LSMs than RRMs, as shown, for example by the global scale Land Data Assimilation System of the NASA Goddard Earth Observing System (Reichle et al., 2014). This platform assimilates simultaneously SMOS soil moisture product, MODIS snow cover extent fraction and integrated GRACE terrestrial water storage variations into an Ensemble Kalman Filter (EnKF) to correct the states of several LSMs. Other studies assimilates similar kinds of observations, along with in situ data, into smaller-scale hydrological models (Trudel et al., 2014). As for RRMs, to the authors knowledge, there are few studies where remotely-sensed and/or in situ data are assimilated into a global-scale RRMs. However, in the literature, there are several studies that used assimilation techniques at smaller and local scale with finer spatial resolution than global RRMs, using mostly in situ data (Schumann and Domeneghetti, 2016). For example, Clark et al. (2008) applied the EnKF to correct soil, aquifer and surface water storage in a small river in New-Zealand (the Wairu). More particularly, they used gauge discharge data from 4 gauges to correct water storages in the 380 sub-catchments dividing the study zone. Paiva et al. (2013a) also used an EnKF over the Amazon basin for three different experiments, which assimilate, first, ENVISAT water surface anomalies from 212 virtual stations, then discharge data from 109 gauges, and finally remotely-sensed discharges from 287 stations obtained from Getirana and Peters-Lidard (2013). This study aimed at correcting discharge estimated by the MGB-IPH hydrological model over more than 5000 sub-catchments composing the Amazon river basin. Moreover, in two different studies, Michailovsky et al. (2013) and Michailovsky and Bauer-Gottwein (2014) assimilated for the Brahmaputra river in Asia and the Zambezi river in Africa, respectively, using an Extended Kalman Filter, ENVISAT water surface elevation measurements from 6 and 9 virtual stations in Michailovsky et al. (2013) and Michailovsky and Bauer-Gottwein (2014), respectively, to correct simulated water volumes in 18 and 37 sub-catchments respectively.

The objective of the present study is to highlight the contribution of a real satellite-based discharge product on a large-scale RRMs. We used an Ensemble Kalman Filter to assimilate discharges derived from ENVISAT water surface elevation measurements. These observations are used to correct the state of the large scale Total Runoff Integrated Pathways (TRIP, Oki and Sud, 1998) RRM version included in the land surface modelling platform "Surfaces Externalisées" (SurfEx, Masson et al., 2013), and developed at the Centre National de Recherches en Météorologie (CNRM, France). This particular version is denoted by CTRIP hereinafter. CTRIP is coupled with the Interactions-Soil-Biosphere-Atmosphere (ISBA, Boone et al., 1999) LSM at a resolution of $0.5° \times 0.5°$.

In section 2, we present the study domain along with the ISBA-CTRIP model version and remotely-sensed product used in this study. Section 3 provides first a general presentation of Ensemble Kalman Filter (EnKF) DA method. Then we introduce the special features associated to the study and the description of the assimilation strategy. Then, in section 4, we present results for a series of DA experiments testing the ensemble generation strategy and the correction of different state variables. Finally, section 5 discusses these results and some perspectives. The last section gives the conclusions and some perspectives of the study.





## 2 Study domain, model and data used

### 2.1 Study domain: the Amazon basin

The study is focused on the Amazon river basin (see Figure 1a). It is the world's largest river in terms of averaged discharge ($2 \times 10^5$ m$^3$ s$^{-1}$) and drainage area ($6.15 \times 10^6$ km$^2$). The discharge at its mouth represents 30% of total freshwater inflow to

the Atlantic Ocean (Wisser et al., 2010) and its catchment area covers about 40% of South America. The river source is located in the Peruvian Andes and it flows through the Brazilian rain forest while receiving water from several important tributaries. First, the Ucayali, the Japurá River, the Purus River and, at Manaus, the Negro River (14% of the total discharge). At this point, the river has reached 56% of its total discharge. From Manaus to its mouth, it receives water from the Madeira River (17% of the total discharge), the Tapajós River and the Xingu River (11% of the total discharge all together) (Molinier et al., 1993).

The Amazon basin's geology can be divided into three major morpho-structural units: the western Andean Cordillera, the central Amazon trough and the shields at the eastern part of the basin (Guiana shield to the North and Brazilian shield to the South). Northern and southern regions of the basin are under a tropical climate with a dry and a wet season, but the maximum rainfall season for the two parts occurs at different periods during the year (Meade et al., 1991). This implies that annual peak discharge in southern tributaries occurs a few months earlier than in northern tributaries. Meanwhile, the central basin is

under an equatorial climate zone, implying high surface temperatures, air humidity and, especially, precipitation. Thus, a vast floodplain along the mainstream is filled every year, leading to the damping of discharge extremes.

### 2.2 ISBA-CTRIP model

#### 2.2.1 Model presentation

The ISBA model (Noilhan and Planton, 1989) is a relatively standard Land Surface Model (LSM) defined over a regular

mesh grid at global scale. The model's equations are solved for each grid cell separately from the others. All grid cells are only correlated through the spatial patterns of atmospheric (especially precipitation) and radiative inputs, vegetation cover and soil composition. By taking into account the heterogeneity in precipitation, topography and vegetation within each grid cell (Decharme and Douville, 2006) and based on the force-restore method (Blackadar, 1976), ISBA gives a diagnosis of the water and energy budgets in each grid cell. Especially, the ISBA-3L configuration (Boone et al., 1999), used in Alkama

et al. (2010) and Decharme et al. (2012), has been chosen for the present study. In this version, the soil is divided into three layers: the superficial layer, the root zone and the sub-root zone. Precipitation can either fall directly on the soil surface or be intercepted by the canopy. The soil water content varies with canopy dripping, surface infiltration, soil evaporation, plant evapotranspiration, surface runoff and deep drainage (for more details, see Alkama et al., 2010; Decharme et al., 2012). Then, ISBA gives a diagnostic of each water budget component, in particular the surface runoff ($Q_{\text{ISBA,sur}}$) and gravitational drainage

($Q_{\text{ISBA,sub}}$) which are the main inputs for CTRIP.

  The CTRIP RRM is also defined over a regular mesh grid. In this study, it is run at the same resolution than ISBA ($0.5° \times 0.5°$). CTRIP is dedicated to the lateral transfer of water from one cell to the other, up to the continent-ocean interface following





a river network (Oki and Sud, 1998). The CTRIP version (Decharme et al., 2010) used in this study is coupled with the ISBA LSM and was subsequently developed by Decharme et al. (2010, 2012). It consists in a system of three reservoirs (see Figure 1b): the surface reservoir $S$ modelling the river, the groundwater reservoir $G$ and the floodplain reservoir $F$, solved using Eq. (1-3) through the estimation of the water mass stored in each reservoir [kg]:

$$\frac{dS(t)}{dt} = Q_{\text{ISBA,sur}}(t) + Q^S_{\text{in,TRIP}}(t) + Q^G_{\text{out}}(t) + Q^F_{\text{out}}(t) - Q^F_{\text{in}}(t) - \frac{v(t)}{L}S(t) \tag{1}$$

$$\frac{dG(t)}{dt} = Q_{\text{ISBA,sub}}(t) - Q^G_{\text{out}}(t) \tag{2}$$

$$\frac{dF(t)}{dt} = Q^F_{\text{in}}(t) - Q^F_{\text{out}}(t) + (P_F(t) - I_F(t) - E_F(t)). \tag{3}$$

Only the surface reservoir $S$ sends water from cell to cell based on the TRIP routing network. A cell can receive water from several upstream cells but sends water into a unique downstream cell based on a space and time-varying flow velocity $v(t)$ estimated with the Manning formula (Manning, 1891). For any given cell, TRIP inputs are the TRIP outflow from upstream cells $Q^S_{\text{in,TRIP}}(t)$ [kg s$^{-1}$] and the ISBA surface runoff for that cell $Q_{\text{ISBA,sur}}$ [kg s$^{-1}$]. Moreover, $S$ receives water from the groundwater reservoir $G$, $Q^G_{\text{out}}(t)$ [kg s$^{-1}$], and can exchange water mass with the floodplain $F$, $Q^F_{\text{out}}(t) - Q^F_{\text{in}}(t)$ [kg s$^{-1}$].

The ISBA gravitational drainage $Q_{\text{ISBA,sub}}$ [kg s$^{-1}$] flows into the groundwater reservoir $G$, whose outflow goes to river reservoir $S$. The groundwater outflow $Q^G_{\text{out}}(t)$ [kg s$^{-1}$] is estimated by

$$Q^G_{\text{out}}(t) = \frac{G(t)}{\tau_G}, \tag{4}$$

where $\tau_G$ [s] is the time delay factor. This simple modelling assumption does not represent the real groundwater dynamics, but only the delay of drainage contribution to the river.

The floodplain scheme activates when the water height in the river, $h_S$ [m], exceeds a given critical bankful height $H_c$ [m]. Then, part of the precipitation is intercepted by the floodplain ($P_F(t)$, [kg s$^{-1}$]) and the water in the floodplain can either evaporates ($E_F(t)$, [kg s$^{-1}$]) or infiltrates in the soil ($I_F(t)$, [kg s$^{-1}$]). A detailed description of the floodplain scheme is given in Decharme et al. (2008, 2010, 2012).

Within a cell, the reservoir $S$ corresponds to an equivalent unique river within the $0.5° \times 0.5°$ cell, that may represent in reality several river branches. The river section is rectangular and is characterized by a slope $s$ [-], a width $W$ [m], a depth $H_c$ [m], a length $L$ [m] and a Manning coefficient $n$ [s.m$^{-1/3}$] that quantifies friction as well as channel resistance at the bottom of the river. A more extensive description of these CTRIP specific parameters is given in Decharme et al. (2012) and Emery et al. (2016).

All these parameters are critical to determine the velocity of surface flow with the Manning formula:

$$v(t) = \frac{s^{\frac{1}{2}}}{n}\left(\frac{Wh_S(t)}{W + 2h_S(t)}\right)^{\frac{2}{3}}, \tag{5}$$

where $h_S$ [m] is the river water height estimated from its actual water mass $S$ and channel geometry.





### 2.2.2 Model implementation over the Amazon

ISBA-CTRIP is run in offline mode. This implies that external atmospheric data are needed to force the model. Here, the atmospheric data from the Global Soil Wetness Projet 3 (GSWP3, http://hydro.iis.u-tokyo.ac.jp/GSWP3) are used. The project consists in three global-scale experiments with the objective of investigating long-term changes of the energy-water-carbon

cycle components and their interactions. The 3-hourly resolution atmospheric boundary conditions used in the present study were generated by dynamically downscaling the global 2°-resolution 20th Century Reanalysis (Compo et al., 2011). This reanalysis assimilates several atmospheric observations into the Climate Forecast System (CFS) operational model from NCEP (National Centers for Environmental Prediction).

For ISBA-CTRIP, the Amazon basin is composed of a total number of 2028 cells. A Sensitivity Analysis (SA) of the ISBA-

CTRIP has been conducted by Emery et al. (2016). In this analysis, the basin was divided into 9 hydro-geomorphological zones which are shown in Figure 2. These zones were designed to take into account different components: (1) hydrological component (the main course is separated from the tributaries which have their own zones); and (2) geological component (the three major morpho-structural units are distinguishable). The 9 zones are the following: (1) the upstream Andean part of the basin until the city of Iquitos, Peru; (2) the main stream from Iquitos to Óbidos; (3) the main stream from Óbidos to the river

mouth; (4) left-bank tributaries from the Napo River to the Japurá River included; (5) left-bank tributaries from the Japurá River to Óbidos including the Negro River and its drainage area; (6) right-bank tributaries from Iquitos to the Purus River confluence at Anamã; (7) right-bank tributaries from Anamã to Óbidos including the Madeira River; (8) right-bank tributaries exiting in zone 3 including the Tapajós River and the Xingu River; and (9) left-bank tributaries exiting in zone 3. This subdivision will be used within the DA platform.

### 2.3 Observations

#### 2.3.1 Altimetry-based discharge product

Altimetry-based discharge product used in this study is derived from water surface elevations measured by the ENVISAT Radar Altimeter-2 altimeter instrument at Virtual Station (VS). VS are computed where the altimeter track crosses the river stream. The ENVISAT mission operated from September 2002 to October 2010 on its nominal orbit, which has a 35 days repeat period

and an 80 km inter-track distance at the equator. The water surface elevations measured over the Amazon basin were initially generated by Silva et al. (2010). The final product was referenced to the EGM2008 geoid (Palvis et al., 2012) and the vertical precision ranged from 12 cm to 30-40 cm for most of the stations (and can reach several meters for the worst stations).

Turning water surface elevation measures into an equivalent discharge requires the use of elevation-discharge rating curves. The rating curves used in this study have been built and validated by Paris et al. (2016), using water surface elevations from

ENVISAT (Silva et al., 2010, 2012, 2014) and discharges simulated by the hydrological-hydrodynamic model MGB-IPH (*Model de Grandes Bacias-Instituto de Pesquisas Hidráulicas*, Collischon et al., 2007). The model original version, developed over the Amazon river basin, consists in a large-scale distributed hydrological model coupled with a hydrodynamic module that uses a simple storage scheme for floodplains (Paiva et al., 2011). The entire basin is divided into 5765 elementary catchments





with an area varying between 100 and 5000 km$^2$. A surface scheme is applied for each mini-basin to estimate the main flows and a routing network is used to direct the flows from one elementary catchment to another, down to the outlet. Two approaches are used to estimate the river discharge: 1- the Muskingum-Cunge method (MC) for basin heads and small tributaries, 2- the Saint-Venant equations (HD for hydrodynamic) for the main stem and main tributaries. The Digital Elevation Model (DEM)

used is SRTM (Farr et al., 2007) and parameters, such as the river width and depth, are determined using geomorphological relationships calibrated over the sub-basins (Paiva et al., 2013a). Moreover, the model version used to determine the rating curves is the version developed by Paiva et al. (2013b), where gauge discharges are assimilated into the model via an Ensemble Kalman Filter (EnKF, Evensen, 2003), over the period between 1998 and 2010. The assimilated discharges allow to correct the simulated discharges over both gauged and ungauged elementary catchments. With a better estimation of discharges, Paiva

et al. (2013b) also provide an estimation on discharge uncertainty (modelled as a white noise) for each elementary catchments.

The MGB-IPH discharges were used by Paris et al. (2016) as baseline to estimate the altimetric rating curves such that:

$$\forall\, j,\, \forall\, i,\, \forall\, t_{\text{alti}},\ Q_{\text{mgb},j(i)}(t_{\text{alti},i}) = a_i \times (H_{\text{alti},i} - z_{0,i})^{b_i}, \tag{6}$$

where

- $H_{\text{alti},i}$ is the altimetric water surface elevation at the $i$-th virtual station which is available at the time $t_{\text{alti},i}$,

- $Q_{\text{mgb},j(i)}(t_{\text{alti},i})$ is the discharge estimated by the MGB-IPH model, at time $t_{\text{alti},i}$, in the $j$-th mini-basin which contains the $i$-th virtual station and

- $a_i$, $z_{0,i}$ and $b_i$ are the rating curve parameters to be determined. Those parameters are constant in time but vary from one virtual station to the other.

To calculate those parameters at each virtual station, a global optimization algorithm, the Shuffled Complex Evolution Metropo-

lis developed by Vrugt et al. (2003), was used. It allowed determining rating curves for 767 ENVISAT virtual stations. More details about the rating curves computation can be found in Paris et al. (2016). Once rating curve parameters are determined, altimetric water surface elevations are easily converted into equivalent "altimetric discharges". Moreover, the altimetric discharges are provided with an estimation of their uncertainty including the normalized deviation from the MGB discharge.

Altimetric discharges has then to be compared to ISBA-CTRIP discharges. However, while the virtual stations are irregularly

distributed over the entire basin, the model is defined over a coarse regular mesh grid of $0.5° \times 0.5°$. A preliminary treatment of the virtual stations is applied to associate each ENVISAT virtual station to an ISBA-CTRIP cell with respect to their localization and the drainage network. The following algorithm has been used:

- The CTRIP river network is compared to a realistic river system (produced with GoogleEarth) to properly associate ISBA-CTRIP cells to a given tributaries in the basin.

- Then, each virtual station is coupled with the closest ISBA-CTRIP cell along the same tributary. It may be the cell containing the virtual station or an adjacent cell according to the river network.





This algorithm allowed associating most of the virtual stations to a unique CTRIP cell. However, some particular cases have been treated. First, some virtual stations were located on tributaries too small to be represented on the TRIP river network. In this case, the virtual station was not included in the study. Then, there were several very close virtual stations associated to the same ISBA-CTRIP cell. In this second case, the virtual stations with the lowest deviation from the MGB discharge were

conserved. Finally, over the 767 ENVISAT virtual stations initially available, 368 ENVISAT virtual stations were kept and associated to an ISBA-CTRIP cell.

### 2.3.2    In situ discharge product

At national or basin scale, water agencies can share discharge time series such as the "Agencia Nacional de Agua" (ANA, hidroweb.ana.gov.br) in Brazil for the Amazon river basin. For the present study, we retrieved discharge time series from 145

ANA in situ stations over the entire basin. These gauge discharges have then been used to evaluate the performances of the DA (but they have not been assimilated in ISBA-CTRIP).

## 3    Method

The purpose of SE DA is to correct model outputs using observations while taking into consideration uncertainties in both the model and the observations. In this study, as observed data correspond to discharge estimates, we chose to correct model

output variables such as discharge or river storage. Indeed, following the results from the ISBA-CTRIP sensitivity analysis (SA; Emery et al., 2016), discharge is mainly sensitive to water inflow. Figure 3 presents the general DA method for the present study. The figure reads from top to bottom and from left to right. Three types of state variables will be considered: the river initial storage, the river final storage (that are both the main ISBA-CTRIP state variables) and the river discharge, that will all be compared to the observed discharge. All three can be corrected through assimilation with specific treatment, that will

be detailed in the following sections. The DA will use several operators (in Figure 3, $\mathcal{M}_{[k-1,k]}$, $\mathcal{Z}_k$ and $\mathcal{H}_k$) that links state variables with each other.

### 3.1    The Ensemble Kalman Filter (EnKF) for ISBA-CTRIP state estimation

The DA technique implemented in the present study is a sequential Ensemble Kalman Filter (EnKF, Evensen, 2003). Here we shortly give the mathematical formalism used in the rest of the paper and a brief description of the EnKF method.

First of all, the assimilation window is the period during which a complete assimilation cycle is conducted. It is delineated by two consecutive observation time and will be denoted by $[k-1,k]$. From now on, the $k$-th assimilation cycle will be the cycle starting at time $k-1$ and assimilating the available observation(s) at time $k$.

### 3.1.1    Control variables

The vector $\mathbf{x}_k \in \mathbb{R}^{n_x}$ is called the control vector. It includes the $n_x$ uncertain variables to be estimated during the $k$-th DA

cycle (within the time interval $[k-1,k]$). As stated before, control variables are prognostic or diagnostic variables of the





ISBA-CTRIP model. Prognostic variables are the physical unknown in the differential equations' system that describes the model's behaviour. Diagnostic variables are also physical variables, but they are estimated from the prognostic variables. The choice of the control variables determines the observation operator $\mathcal{H}_k$ that maps the control variables into the observation space:

$$\mathbf{y}_k = \mathcal{H}_k(\mathbf{x}_k),\tag{7}$$

where $\mathbf{y}_k$ are the control variables equivalent in the observation space also called model observations. They are then compared to the measured observations $\mathbf{y}_k^o$ (described in Section 3.1.2) during the DA step.

Unlike hydrodynamic models, which directly solve Saint-Venant equations and for which discharge is a model state variable (or prognostic variable), the hydrological model ISBA-CTRIP solve differential equations describing the time evolution of
water stock in the river ($S$), the groundwater ($G$) and the floodplain ($F$). Then, water elevation and river discharge are diagnostic variables derived from these prognostic variables. In CTRIP, river discharge $Q_{\text{out}}^S$ is computed as follows:

$$Q_{\text{out}}^S = L^{-1}vS \quad [\text{kg.s}^{-1}],\tag{8}$$

with $L$ [m] river section length, $v$ the flow velocity (estimated from the Manning formula) and $S$ the surface water mass.

Therefore, three types of variables can be considered as control variables in the data assimilation scheme: the discharge $Q_{\text{out}}^S$
(denoted $Q$ in the remaining of the study to simplify notation, which is a diagnostic variable), the river final water stock $S_{\text{end}}$ (a prognostic variable) or the river initial water stock 5also a prognostic variable). Definition and complexity of the observation operator $\mathcal{H}_k$, that maps the control space into the observation space, depends on the nature of the control variable. These three options are presented below.

- *Option 1: correcting ISBA-CTRIP discharges*

For this option, the control variables, gathered into the vector $\mathbf{x}_k$ are the ISBA-CTRIP discharges $Q_{i,k}$, $i = 1 \ldots n_x = 2028$ (number of TRIP cells in the Amazon basin) estimated for each 2028 cells in the TRIP Amazon basin, at the assimilation cycle $k$.

The observation operator $\mathcal{H}_k$ resumes to a selection operator $\mathcal{S}_k$ which select the observed TRIP cells at the current assimilation cycle:

$$\mathcal{H}_k = \mathcal{S}_k.\tag{9}$$

This operator is linear. The difficulty with this operator is that, once the assimilation analysis is produced, it is necessary to convert the analysis discharge $Q_{i,k}^a$, $i = 1 \ldots n_x$ (i.e. corrected discharge obtained after assimilation), into the equivalent final water stock $S_{\text{end},i,k}^a$. Indeed, as already stated, in ISBA-CTRIP, discharge is not a pronostic variable. Correction from the assimilation step needs to be propagated to the model prognostic variables, here, the river final stock. Moreover,
the analysis final water stock $S_{\text{end},i,k}^a$ will be used as initial condition for the model run until the next assimilation cycle: $\forall i = 1 \ldots n_x,\ \forall k,\ S_{\text{end},i,k}^a = S_{\text{ini},i,k+1}^b$. Yet, the exact relationship linking discharge to the final river stock is unknown.





A possible solution consists in inverting Eq. 8. Assuming that the discharge estimated by ISBA-CTRIP $Q_{i,k}^a$ is the instantaneous flow at the final time of the integration window:

$$Q_{k,i,[\mathrm{kg.s}^{-1}]}^a = L^{-1} v S_{\mathrm{end},k,i,[\mathrm{kg}]}^a \iff Q_{k,[\mathrm{m}^3.\mathrm{s}^{-1}]}^a = \rho^{-1} L^{-1} v S_{\mathrm{end},k,i,[\mathrm{kg}]}^a.$$

We obtain that (for more details on this approximation, see appendix A):

$$S_{\mathrm{end},k,i,[\mathrm{kg}]}^a \approx \rho L W^{2/5} s^{-3/10} n^{3/5} \left( Q_{k,i,[\mathrm{kg.s}^{-1}]}^a \right)^{3/5}, \tag{10}$$

with $\rho$ [m$^3$.kg$^{-1}$] the water density, $L$ [m] the river section length, $W$ [m] the river width, $s$ [-] the riverbed slope and $n$ [-] the Manning coefficient in the riverbed. Then, for experiments with discharges as control variables, the formula in Eq. 3.1.1 will be used to convert corrected discharges into river stock and then propagate the model to the next observation time.

- *Option 2: correcting ISBA-CTRIP final water stock*

For this option, the control variables, gathered into the vector $\mathbf{x}_k$ are the ISBA-CTRIP final water stock $S_{\mathrm{end},i,k}$, $i = 1 \ldots n_x$ estimated for each 2028 cells in the TRIP Amazon basin, at the assimilation cycle $k$.

The computational cost for this option is the same as for the first option but, now, the observation operator is defined as

$$\mathcal{H}_k = \mathcal{S}_k \circ \mathcal{Z}_k, \tag{11}$$

where $\mathcal{Z}_k$, is the operator (implicitly defined within TRIP) that turns the river final stock $S_{\mathrm{end},i,k}$ into equivalent discharge $Q_{i,k}$. Even though $\mathcal{H}_k$ is not linear any more, this option presents the advantage of correcting the river final stock $S_{k,\mathrm{end}}^a$ that can be directly used for the next assimilation cycle and no additional uncertainties are introduced. However, the corresponding analysis discharge $Q_{i,k}^a$ is now unknown as the explicit expression of $\mathcal{Z}_k$ is also unknown. A potential formula to determine $Q_{i,k}^a$ can be deduced from Eq. 3.1.1. Such a formula will be necessary to make comparative statistics to ENVISAT and in situ discharges and be able to evaluate the assimilation performances.

- *Option 3: correcting ISBA-CTRIP initial water stock*

For this final option, the control variables, gathered into the vector $\mathbf{x}_k$ are the ISBA-CTRIP initial water stock $S_{\mathrm{ini},i,k}$, $i = 1 \ldots n_x$ estimated for each 2028 cells in the TRIP Amazon basin, at the assimilation cycle $k$.

The discharge observations are used to correct the surface water stock at the time prior to the observation time or, in other words, at the initial time of the integrating window. Therefore, the observation operator is written as the composition of the model operator $\mathcal{M}_{[k-1,k]}$ with the observation operator defined in Eq. 11:

$$\mathcal{H}_k = \mathcal{S}_k \circ \mathcal{Z}_k \circ \mathcal{M}_{[k-1,k]}. \tag{12}$$

This operator is highly non-linear as it contains the full model operator. However, it is the only option where no uncertainties are added from the use of an external formula to compute corrected discharge at the observation time. Uncertainties





in the stock-discharge relationship are only due to the model uncertainties. Indeed, once the analysis initial water stock $S^a_{\mathrm{ini},k,i}$ is determined, the control variables update must be propagated through the next assimilation time by re-integrating the ISBA-CTRIP model over the assimilation window. The initial water stock $S^a_{\mathrm{end},k,i}$ and the analysis discharge $Q^a_{k,i}$ are automatically determined during this run.

### 3.1.2   Observation variables

In the framework of the state estimation, the observation variables, at a given day within the Amazon basin, are the discharge estimates derived from ENVISAT water surface elevations at the virtual stations associated to an ISBA-CTRIP cell. The ENVISAT repeatability is 35 days, therefore a given virtual station will provide an observation every 35 days at best. During the data assimilation experiments, all virtual stations will be used simultaneously. Because of the ENVISAT orbit, the number

of available observations at a given day will vary between 0 and 15, and these observations will be assimilated daily via the EnKF. Then, the observation vector $\mathbf{y}^o_k$ at the assimilation cycle $k$ (equivalently, at the simulation day $k$) is composed of the $n_{y,k}$ discharge measures available at day $k$:

$$\mathbf{y}^o_k = \left[ y^o_{k,1}, \quad y^o_{k,2}, \quad \ldots, \quad y^o_{k,n_{y,k}}, \right] \tag{13}$$

where $y^o_{k,j}$, $j = 1 \ldots n_{y,k}$ is the $j$-th observation among the $n_{y,k}$ at cycle $k$.

Measurement errors $\epsilon^{\mathbf{m}}_{\mathbf{k}}$ come from errors in ENVISAT water surface elevations, errors in MGB discharges and uncertainties in the rating curves parameters used to turn water surface elevation into discharge. Sorooshian and Dracup (1980), Clark et al. (2008) and Paris et al. (2016) noticed that the concavity of the elevation-discharge relationship implies that the higher a water elevation, is the more uncertain the corresponding discharge. Therefore, the observation error standard deviation $\sigma^o_{k,j}$, associated to the $j$-th observation at cycle $k$, is defined with respect to the instantaneous discharge measure $y^o_{k,j}$, i.e :

$$\sigma^o_{k,j} = \eta^o_j \times y^o_{k,j}, \; j = 1 \ldots n_{y,k} \tag{14}$$

where $\eta^o_j \in \, ]0,1[$ is a constant depending on the virtual station, such that $\eta^o_j$ models the relative error. The observation error standard deviation $\sigma^o_{k,j}$ is then a fraction of the instantaneous discharge. For each virtual station, the value of $\eta^o_j$ depends on, first, the deviation from the MGB discharge, noted $\sigma^{\mathrm{mgb}}_j$ [%] and determined by Paris et al. (2016). As MGB discharges were used to determine ENVISAT discharges from ENVISAT water elevations, $\sigma^{\mathrm{mgb}}_j$ represents the deviation between the two

discharges data. Second, to take into account that MGB discharge is not perfect (in other words, to take into account some deviation from the real discharge), 0.05 is added to $\sigma^{\mathrm{mgb}}_j$ and the sum is rounding up to the nearest whole number, giving

$$\eta^{\mathrm{mgb}}_j = \mathrm{E}\left( \sigma^{\mathrm{mgb}}_j + 0.05 \right),$$

where the function E is the ceiling function. Finally, $\eta^o_j$ is equal to the first multiple of 5, above $\eta^{\mathrm{mgb}}_j$. At the end, $\eta^o_j$ varies from 0.10 to 0.35 over the entire basin and is constant in time. Besides, the representativeness error $\epsilon^{\mathbf{r}}_{\mathbf{k}}$ induced when a virtual

station is associated to cell of the coarse TRIP mesh grid is neglected here.



Moreover, for a given assimilation cycle and also between different cycles, the observations are considered uncorrelated in space and time. The observation error covariance matrix at cycle $k$ $\mathbf{R}_k$ is then a diagonal definite positive square matrix such that:

$$
\mathbf{R}_k = \begin{bmatrix} (\sigma_{k,1}^o)^2 & 0 & \dots & 0 \\ 0 & (\sigma_{k,2}^o)^2 & \dots & 0 \\ \vdots & \vdots & \ddots & \vdots \\ 0 & \dots & 0 & (\sigma_{k,n_{y,k}}^o)^2 \end{bmatrix}. \tag{15}
$$

### 3.1.3 The EnKF sequential estimation

The Kalman Filter (KF) and its extensions (such as the EnKF) is divided in two steps:

1. *The prediction step*, during which the modelled system is propagated in time over the assimilation window $[k-1,k]$ through an integration of the ISBA-CTRIP model $\mathcal{M}_{[k-1,k]}$ and given the uncertainties in the system.

2. *The analysis step*, in which new available observations at time $k$, $\mathbf{y}_k^o$, are taken into consideration to consistently update the control variable.

Under linear assumptions for the model and observation operator, the KF analysis estimate resumes to:

$$
\mathbf{x}_k^a = \mathbf{x}_k^b + \mathbf{K}_k \left( y_k^o - \mathbf{H}_k \mathbf{x}_k^b \right), \tag{16}
$$

$$
\mathbf{K}_k = \mathbf{P}_k^b \mathbf{H}_k^T \left( \mathbf{H}_k \mathbf{P}_k^b \mathbf{H}_k^T + \mathbf{R}_k \right)^{-1}, \tag{17}
$$

where $\mathbf{H}_k$ is the observation operator linear tangent but, under linear assumptions, is equal to the observation operator. $\mathbf{K}_k$ is the Kalman gain matrix defined from the background and observation error covariance matrices $\mathbf{P}_k^b$ and $\mathbf{R}_k$ (supposed known).

In the EnKF framework, the model and observation operator are not linear. Also, the control error covariance matrix is unknown. Therefore, the main idea is to use stochastic ensembles to represent the control variables PDFs along with the error models (Evensen, 1994, 2003). So, firstly, the background control variables $\mathbf{x}_k^b$ are stochastically represented by an ensemble of $n_e$ members:

$$
\mathbf{X}_{e,k}^b = \begin{bmatrix} \mathbf{x}_k^{b,[1]} & \mathbf{x}_k^{b,[2]} & \dots & \mathbf{x}_k^{b,[n_e]} \end{bmatrix}, \tag{18}
$$

Each member is estimated separately through the EnKF prediction step. For each control variables case (see Section 3.1.1), each member of the control ensemble $\mathbf{X}_{e,k}^b$ are estimated by integrating the model operator from the corresponding analysis member at the previous assimilation cycle, while adding external uncertainties (see Section 3.2):

$$
\forall \, l = 1 \dots n_e, \ \mathbf{x}_k^{b,[l]} = \mathcal{M}_{[k-1,k]}(\mathbf{x}_{k-1}^{a,[l]}). \tag{19}
$$

Then, the background control ensemble must be compared to the observations. Depending on the control variables nature, the model operator is already included (option 3) or not (option 1 and 2) within the observation operator. Besides, following





Burgers et al. (1998), an additional noise $\epsilon_k^o$ is added to the observation vector $\mathbf{y}_k^o$ to avoid ensemble collapse. The observation ensemble thus obtained is noted:

$$\mathbf{Y}_{e,k}^o = \begin{bmatrix} \mathbf{y}_k^{o,[1]} & \mathbf{y}_k^{o,[2]} & \cdots & \mathbf{y}_k^{o,[n_e]} \end{bmatrix}. \tag{20}$$

Finally, the EnKF analysis step (see Eq. 16) is applied to each member of the ensemble such that

$$\forall\, l = 1\ldots n_e,\ \mathbf{x}_k^{a,[l]} = \mathbf{x}_k^{b,[l]} + \mathbf{K}_{k,e}\left(\mathbf{y}_k^{o,[l]} - \mathcal{H}_k(\mathbf{x}_k^{b,[l]})\right), \tag{21}$$

where the direct non-linear observation operator is applied to convert the control variables into equivalent model observations.

The other particularity of the EnKF is that the Kalman gain is also stochastically estimated from the different control and model observation ensemble such that:

$$\mathbf{K}_{e,k} = [\mathbf{PH}^T]_{e,k}\left([\mathbf{HPH}^T]_{e,k} + \mathbf{R}_k\right)^{-1}. \tag{22}$$

## 3.2 Generating the ensembles

The background error covariance matrices $[\mathbf{PH}^T]_{e,k}$ et $[\mathbf{HPH}^T]_{e,k}$ are estimated from the definition suggested by Evensen (2004); Moradkhani et al. (2005); Durand et al. (2008). Details on how they are exactly calculated are given in Appendix B. These matrices have a $n_x \times n_{y,k}$ and $n_{y,k} \times n_{y,k}$ size, respectively. The elements in the error covariance matrices, depend only on the definition of the background ensemble stored in the control matrix $\mathbf{X}_{e,k}^b$ and the parameter uncertainties taken into consideration to generate $\mathcal{H}(\mathbf{X}_{e,k}^b)$. In the framework of state estimation, we choose to consider uncertainty into the initial condition and uncertainty into the precipitation forcing (Nijssen and Lettenmaier, 2004; Andreadis and Lettenmaier, 2006; Clark et al., 2008; Paiva et al., 2013b).

### 3.2.1 General definition of the error covariance matrices

### 3.2.2 Perturbation of the initial condition

The vector containing initial surface reservoir storage for each $n_x = 2028$ CTRIP cell at the assimilation cycle $k$ is called $\mathbf{c}_k$. To ease the notations, we will omit, as much as possible, the assimilation cycle $k$ subscript, knowing that, for all randomly perturbed variables and constants, a new ensemble is generated at each cycle.

We used the Amazon basin division into $n_s = 9$ hydrogeomorphological zones (see Figure 2). Initial conditions are perturbed by applying a multiplying factor over each zone $\eta_s^{\mathbf{c},[l]}$ such that

$$\forall\, s = 1\ldots n_s,\ \forall\, l = 1\ldots n_e,\ \mathbf{c}_{s,k}^{[l]} = \eta_i^{\mathbf{c},[l]}.\mathbf{c}_{s,k}, \tag{23}$$

where $\mathbf{c}_{s,k}$ is the reduction of the initial condition $\mathbf{c}_k$ to the only zone $s$. For the perturbation to vary from one member to another, the value $\eta_s^{\mathbf{c},[l]}$ is the realization of a Gaussian distribution, different for each member $[l] = 1\ldots n_e$ and for each hydrogeomorphological zone $s$. The Gaussian distributions used have a mean value of 1 and a standard deviation $\sigma_{\eta_{s,k}^{\mathbf{c}}}$, which values are detailed in Table 1.

The $\eta_i^{\mathbf{c},[l]}$ values depend on the assimilation cycle $k$ and on the hydrogeomorphological zone in which the $i$-th cell is.



- Firstly, a more important perturbation is applied to cells situated on the river mainstream (zone 2 and 3) as we assume that the uncertainties are more important in those zones. Indeed, discharges in these zones are the highest of the entire basin. Besides, several cells are confluence cells and are subject to backwater effects. As ISBA-CTRIP does not model the backwater effects, the water stock uncertainties in these celles are increased.

- Secondly, at the first assimilation cycle, the initial condition before perturbation $\mathbf{c}_1$ is identical for every member. At this particular cycle, the ensemble variance after perturbation depends only on the perturbation method presented in Eq. 23 while, for the other assimilation cycles, the successive previous assimilation cycles introduced an additional variability between members, before the perturbation step in Eq. 23. Therefore, the initial condition variance is more important at the second assimilation cycle and after. Then, to generate a larger variability at the first assimilation cycle, the standard

deviation $\sigma_{\eta^{\mathbf{c}}_{s,k}}$ of the variable $\eta_i^{\mathbf{c},[l]}$ is more important at the first cycle than for the others.

### 3.2.3   Perturbation of precipitations

Another source of uncertainties for the generation of the ensemble $\mathcal{H}_k(\mathbf{X}_e^b)$ lies in the precipitation fields. Atmospheric forcing come from GSWP3 product (see Section 2.2.2). Precipitation forcing $\mathbf{F}$ have been perturbed using presented by Clark et al. (2008). The ensemble of perturbed precipitation fields $\widetilde{\mathbf{F}}_e$ is defined such that:

$$\widetilde{\mathbf{F}}_e = \left\{ \widetilde{\mathbf{F}}^{[1]}, \quad \widetilde{\mathbf{F}}^{[2]}, \quad \ldots \quad \widetilde{\mathbf{F}}^{[n_e]} \right\} = \left\{ \varphi_p^{[1]}.\mathbf{F}, \quad \varphi_p^{[2]}.\mathbf{F}, \quad \ldots \quad \varphi_p^{[n_e]}.\mathbf{F} \right\}, \tag{24}$$

where:

- $\mathbf{F}$ is the two-dimensional field of precipitation forcing before pertubation (with a time-step of 3 hours),

- $\widetilde{\mathbf{F}}^{[l]}$, for $l = 1 \ldots n_e$, is the $l$-th perturbed precipitation field,

- $\varphi_p^{[l]}$, for $l = 1 \ldots n_e$, is the $l$-th multiplying uniformly-distributed field of $\mathbf{F}$ to generate $\widetilde{\mathbf{F}}^{[l]}$. More details on how the

fields $\varphi_p^{[l]}$ have been generated are given in Appendix C.

### 3.2.4   Localization of the error covariance matrices

In the framework of state estimation, the sampling error can introduce artificial correlations into the background/analysis error covariance matrices, and generate spurious correlations between two distant grid cells in the mesh (Anderson, 2007). The ensemble size $n_e$ is limited by computational constraints. Therefore, before the EnKF analysis step, a numerical processing of

the matrices $[\mathbf{PH}^T]_{e,k}$ and $[\mathbf{HPH}^T]_{e,k}$ matrices is necessary to suppress spurious correlations that can potentially degrade the analysis. Localization methods are designed to reduce these problems.

    A first classical localization technique consists in explicitly modifying the background error covariance matrix $\mathbf{P}^b_{e,k}$ by nullifying correlations between two cells that are far away from each other (Houtekamer and Mitchell, 1998; Anderson, 2001). Another technique, more indirect, consists in multiplying the matrix $\mathbf{P}^b_{e,k}$ by a correlation matrix generated from a radial

function, namely a function of the two/three spatial dimensions which monotonously decreases with the distance between





control variables (Hamill et al., 2001; Houtekamer and Mitchell, 2001, 2005). A sparse matrix $\tilde{\mathbf{P}}^b_{e,k}$ is therefore computed, with non-zero elements centred around the matrix diagonal. This modified matrix replaces $\mathbf{P}^b_{e,k}$ in the calculation of the Kalman gain matrix $\mathbf{K}_{e,k}$. Among other localization techniques, the LETKF (Local Ensemble Transform Kalman Filter, Hunt et al., 2007) combines the classical EnKF with a localization technique which consists in proceeding the analysis step into
characteristic sub-spaces of the overall problem space.

However, all these localization techniques described above have been developed for atmospheric modelling where problems are in two or three dimension. The use of localization in hydrology is more limited. Several studies exist to improve subsurface flow modelling (Devegowda et al., 2010; Delijani et al., 2014) but these approaches have a dimensionality close to atmospheric approaches as they take place in a continuous medium in two or three dimensions. Other studies using localization allow
estimating better model parameters, still continuously defined in two or three dimensions (Sun et al., 2009; Rasmussen et al., 2015).

The localization method used with the CTRIP river routing model can not be simply defined on a two-dimensional radial function. Indeed, the river flow is along several one-dimensional flow directions, modelled by the routing network. The localization technique must consider the routing network to decorrelate adjacent cells on the mesh grid but located in two different
sub-catchments. Nevertheless, along a same flow direction, the correlation between two distinct cells depends on the distance between the two cells. Then, for each assimilation cycle, the localization consists in a localization mask delimiting an influence area for each observation. These influence areas gather a limited number of neighbouring downstream and upstream cells around the observed cell with respect to the river routing network.

To determine the number of cells defining the influence area, the basin subdivision in 9 hydro-geomorphological zones is
used with a mean flow velocity for each zone. The influence area, for a given observed cell, is given by the criteria below. For an influence area of size $p$ cells, the area is composed of:

- the observed cell,

- the $p$ downstream cells according to the river routing network,

- all the cells upstream the observed one covering $p$ upstream levels. However, the going up stops when the hydro-
geomorphological zone is different from the one of the observed cell.

The number of cells within the influence area depends on the mean flow velocities (averaged over a year of simulation) in the zone in which the considered cell is situated. Those mean velocities are calculated from the free run simulation, namely the ISBA-CTRIP simulation realized without any assimilation step. The ISBA-CTRIP resolution is $0.5° \times 0.5°$, or approximately $50\mathrm{km} \times 50\mathrm{km}$. Given the river meanderings, the minimal covered distance through a cell is of 50 km. Furthermore, by comparing
the free run discharge to in situ and ENVISAT discharges, it seems that the free run underestimate discharge (and so the flow velocity). Consequently, to fix the number $p$ of cells defining the influence area in each hydro-geomorphological zone, the following steps have been performed:

1. the mean velocity for the cells into a given zone is converted into an equivalent distance in km,





2. the maximal distance within the zone is kept and rounded to the closest higher multiple of 50,

3. the number $p$ determining the size of the influence area is the number of cells covered by the maximal rounded distance, knowing that 50 km = 1 cell.

The number of cells into the influence area is presented, for each zone, in the Table 2. The final localization mask is presented

into a matrix of size $n_x \times n_{y,k}$ (with $n_x$, the number of control variables, and $n_{y,k}$, the number of observation variables, at the assimilation cycle $k$) containing only 0 and 1. The localization mask has the same dimension as the matrix $[\mathbf{PH}^T]_{e,k}$. So, to apply the localization, the localization matrix is term-to-term multiplied to the error covariance matrix $[\mathbf{PH}^T]_{e,k}$. To then extend the localization to the error covariance matrix $[\mathbf{HPH}^T]_{e,k}$, the lines in $[\mathbf{PH}^T]_{e,k}$ corresponding to the observed cells are extracted to form the second localization matrix. This second matrix is also term-to-term multiplied to $[\mathbf{HPH}^T]_{e,k}$. This

localization step is applied in each assimilation cycle with respect to the activated ENVISAT virtual stations at the current assimilation cycle.

### 3.3 Assimilation diagnostics

During the assimilation experiment, it is necessary to quantify the assimilation performances. The quality of the assimilation will be evaluated in a given cell $i$ by estimating the Root Mean Square Error (RMSE) between the simulated discharge $Q_i^*$ and

the observed discharge $Q_i^\dagger$, giving:

$$\mathrm{RMSE}_i^{*,\dagger} = \sqrt{\frac{1}{K^\dagger} \sum_{k=1}^{K^\dagger} \left( Q_{i,k}^* - Q_{i,k}^\dagger \right)^2} \quad \left[ \mathrm{m}^3.s^{-1} \right]. \tag{25}$$

$K^\dagger$ represents the total number of available observed discharge $Q_i^\dagger$ at the studied cell for the study period. The "$*$" upperscript can be either the upperscript "$f$" for the free run (without assimilation) or the upperscript "$a$" for the analysis run (after assimilation). The "$\dagger$" upperscript can be either "$o$" for the observed ENVISAT discharge or "$situ$" for the gauge discharge.

Based on this definition, the assimilation performance will be estimated at each cell with the normalized RMSE (RMSEn) defined by:

$$\mathrm{RMSEn}_i^{*,\dagger} = 100 \times \frac{\mathrm{RMSE}_i^{*,\dagger}}{\overline{Q_{i,\bullet}^\dagger}}, \quad [-], \tag{26}$$

where $\overline{Q_{i,\bullet}^\dagger}$ corresponds to the mean of $Q_{i,k}^\dagger$ averaged over the available time steps $k$.

Also, to evaluate the global performance of the assimilation over the entire basin, a global RMSEn (RMSEn$_{\mathrm{global}}$) will be

determined by:

$$\mathrm{RMSEn}_{\mathrm{global}}^{*,\dagger} = 100 \times \frac{1}{\left( \sum_{i=1}^{I^\dagger} w_i \right)} \sum_{i=1}^{I^\dagger} w_i.\mathrm{RMSEn}_i^\dagger, \quad [-], \tag{27}$$



where $I^\dagger$ is the total number of stations and $w_i$ a weighting constant depending on the maximal discharge at the $i$-th station ($\max_k \left( Q_{i,\cdot}^\dagger \right)$) and the maximal discharge over the basin ($\max_{i,k} \left( Q_{\cdot,\cdot}^\dagger \right)$) such that:

$$w_i = \frac{\log_{10}\left[ \max_k \left( Q_{i,\cdot}^\dagger \right) \right]}{\log_{10}\left[ \max_{i,k} \left( Q_{\cdot,\cdot}^\dagger \right) \right]}. \tag{28}$$

With this weighting, cells along the mainstream and the largest tributaries (with the highest discharges) have more importance in the global statistics than cells located in basin heads.

Besides, the analysis run is available as an ensemble. The statistics will then be estimated for each member of the ensemble and the mean (see Eq. 29) of the ensemble will be presented such as:

$$\overline{\mathrm{RMSEn}_i^{a,\dagger}} = \frac{1}{n_e} \sum_{l=1}^{n_e} \mathrm{RMSEn}_i^{a,[l],\dagger}, \tag{29}$$

where $\mathrm{RMSEn}_i^{a,[l],\dagger}$ is the normalized Root Mean Square Deviation of the $l$-th member of the analysis discharge ensemble.

## 3.4 Assimilation strategy

The state estimation experiments have the objective to test the different control variables described in Sections 3.1.1. Also, another objective is to test, validate and criticize the localization mask introduced in Section 3.2.4. In the following, experiment names using the localization will have the "-local" suffix, the one without any localization will have the "-direct" suffix and the one with no correlation between cells will have the "-diag" suffix. The objective of this study is to determine the best strategy to assimilate ENVISAT discharges into the ISBA-CTRIP model using the EnKF to correct the model state variables. The different experiments are presented in Table 3. After analysing these 5 elementary simulations over a single year, a last experiment will run over the entire ENVISAT observing period (from September 2002 to June 2010), based on the best configurations.

For all the DA experiments, the observation errors are those described in Section 3.1.2, the model errors are those presented in Sections 3.2.2 and 3.2.3. Moreover, each experiment in Table 3 lasts 365 assimilation cycles of 1 day (so 1 year of assimilation) from January 1$^{\text{st}}$, 2009 to December 31$^{\text{st}}$, 2009. We chose this period as it overlaps with other altimetry mission (namely JASON-2) and future works may include comparing the two dataset contribution. The numerous ensemble ISBA-CTRIP simulations were realized with the High Performance Computation Platform CALMIP (https://www.calmip.univ-toulouse.fr/spip/) with the supercomputer EOS.

In SE1-direct experiment, ENVISAT discharges are assimilated to correct the initial surface reservoir storage in TRIP (and inherently TRIP simulated discharges). For this first experiment, a classical EnKF, without any localization treatment of the error covariance matrices $[\mathbf{PH}^T]_{e,k}$ and $[\mathbf{HPH}^T]_{e,k}$, is conducted. This first experiment will be compared to the two next experiments, SE1-diag and SE1-local, which will highlight the contributions and /or limitations of the chosen localization approach. Finally, the two last experiments, SE2-local and SE3-local, will test the other possible control variables and the reliability of the operator $\mathcal{Z}_k$. More particularly, the experiment SE2-local is based on the control vector option 2 (see Section 3.1.1) that assimilates discharges to correct the final rive storage, and SE3-local is based on the control vector option 1 (see Section 3.1.1) that assimilates discharges to directly correct the ISBA-CTRIP discharges.





## 4 Results

### 4.1 Free run performances

The current section briefly presents the model performance without assimilation called the free run. As all in and ENVISAT VS have been associated to a unique ISBA-CTRIP cell, it is possible to compare observed discharge at these stations to corresponding ISBA-CTRIP simulated discharge. To begin with, a sample of 12 in situ stations, spread over the entire basin (over the mainstream and the main tributaries) is selected. The location and the name of these stations is represented in Figure 4. Figure 5 compares ISBA-CTRIP free run discharges to in situ and ENVISAT discharges at the 12 stations over one year of simulation (year 2009). From this comparison the following observations can be drawn:

- over the majority of cells where there are both an in situ and a virtual station, the two discharge time series are similar (but not identical, see Figure 5 1-3,5-6,8-9,12). These results are due to the fact that gauge discharges where directly assimilated into the MGB-IPH hydrological model to correct the MGB-IPH estimated discharges (Paiva et al., 2013a). Then, those same estimated discharges were used to calculate parameters of the rating curves between ENVISAT water elevations and MGB discharges (Paris et al., 2016). Even though these rating curves have been derived from a model that assimilated in situ data, ENVISAT-derived discharges depend essentially on the remotely-sensed water surface elevations variations (Paris et al., 2016). Therefore, ENVISAT discharges remain independent enough from in situ data.

- A strong difference between the in situ and ENVISAT discharges could indicate either that the rating curve parameters were not correctly estimated or that in situ/ENVISAT/MGB-IPH discharges have strong errors. As an example, see Figure 5 11 at Itaituba where the gauge discharge is discontinuous and is even equal to 0 for some dates. Another example is the gauge discharge at Manicoré, in Figure 5 10 (Paris et al., 2016).

- Finally, in most cases, the free run discharge is quite different from the observed discharge. At downstream mainstream stations (at Manacapuru and Óbidos in Figure 5 panels 2 and 3), the ISBA-CTRIP model is not able to reproduce flooding occurring between June to August. Therefore, in the free run, the discharge peak occurs earlier in the year and the discharge variations in this period are faster than the observed discharge variations. Similarly, at most of right-bank tributary stations, the free run discharge peak is higher than the observed discharge peak (see Figure 5 7-12). However, the seasonal cycle is well-reproduced for all these stations. These results illustrate the necessity to conduct the DA experiments.

Then, Figure 6 displays the global performances of the free run. For each ENVISAT virtual stations (see Figure 6a) and each in situ stations (see Figure 6b), the RMSEn (defined in Eq. 26) between the simulated and the observed discharges is calculated and its value is indicated by a colour at the location of the station over the basin. The results are similar between ENVISAT and gauge discharges, confirming good concordance between the two discharges data sets. RMSEn show important deviations in basin heads on most of the tributaries as well as at confluence between right-bank tributary and mainstream. Apart from confluence and basin heads, the largest tributaries, such as the Negro and the Madeira, are well represented. Concerning global





statistics (see Eq. 27), $\text{RMSE}_{\text{global}}^{f,o}$ is equal to $71.12\%$ compared to ENVISAT discharges and $\text{RMSE}_{\text{global}}^{f,simu}$ is equal to $68.96\%$ compared to gauge discharges. These deviations are likely due to atmospheric forcing, parametrization and modelling errors, especially floodplains parametrization. The DA experiments will focus on correcting the model outputs which result from those uncertainties.

## 4.2 Evaluation of the localization method

The first series of experiments assimilates ENVISAT discharges to correct the ISBA-CTRIP initial river stock (see the three first rows in Table 3). They differ on the definition of the background error covariance matrices $[\mathbf{PH}^T]_{e,k}$ and $[\mathbf{HPH}^T]_{e,k}$. The experiment SE1-direct uses the complete stochastic matrix defined in Eq. B1 and B2. In experiment SE1-diag, these matrices are processed such that covariance between two different CTRIP cells is set to $0$ if the two variables are situated in two different CTRIP cells. Lastly, SE1-local is based on the localized version of the matrices presented in section 3.2.4. So, Table 4 displays the global RMSEn (see definition in Eq. 27) for the three experiments compared to the free run global statistics. From Table 4, we can see that the RMSE between the free run discharge and both the ENVISAT and the gauge discharge is reduced for all experiments, showing that the data assimilation platform is working correctly. The SE1-diag experiment gives the worst results when compared to both the ENVISAT discharge and the gauge discharge. Then, compared to ENVISAT discharges, SE1-local gives the best results by reducing the global RMSEn of more than $56\%$ ($49\%$ for SE1-direct) while SE1-direct presents slightly better global statistics than SE1-local when compared to gauge discharges (RMSEn are reduces by $16.5\%$ for SE1-direct and $15.25\%$ for SE1-local). Overall, the global statistics are more reduced when compared to ENVISAT discharges than to gauge discharges. This is due to the fact that gauge discharges are not directly assimilated, unlike ENVISAT discharges. The next subsections present and analyse in more details results from each experiment.

### 4.2.1 SE1-direct results

Figure 7 displays the mean analysis RMSEn ($\overline{\text{RMSEn}_i^{a,\dagger}}$ defined in Eq. 29) for each ENVISAT virtual stations (Figure 7a) and for each in situ stations (Figure 7b). First of all, results between the ENVISAT RMSEn and in situ RMSEn are similar, due to the similarity between ENVISAT and gauge discharge time series at most stations. According to Figure 7a, the assimilation worked quite well along the mainstream and the main left-bank tributaries, namely the Negro river, the Japurá and the Içá, with several stations where RMSEn are below $20\%$. The assimilation performances are more moderate over right-bank tributaries where RMSEn are mostly between $20\%$ and $60\%$. Over the entire basin, RMSEn remain high in all basin heads, along small tributaries and also at most confluences, see for example the Jutaí-Solimões confluence (RMSEn above $60\%$), Purus-Solimões/Madeira-Solimões/Tapajós-Amazon confluences (RMSEn above $40\%$) or Xingu-Amazon confluence (RMSEn above $80\%$).

Figure 8 compares the mean analysis discharge in red line at the 12 stations previously introduced in section 4.1. For most stations, we can see that the mean analysis discharges recovers a seasonal cycle closer to the observations than the free run. It is especially true for stations along the mainstream, namely Sao Paulo de Olivenca, Manacapuru and Óbidos (Figure 8, panels 1-3). Also, for stations along right-bank tributaries (Figure 8, panels 7-12), the analysis seasonal discharge peak is lowered





compared to the free run seasonal discharge peak and fits better the observations. This shows the good functioning of the assimilation platform.

Nevertheless, mean analysis discharge for all displayed stations presents a chaotic behaviour with numerous local minima and maxima. We can assume that this behaviour is present for all CTRIP cells in the basin. Moreover, for a given cell, most
of these sudden variations are asynchronous with ENVISAT observation dates for this cell. For example, at Serrinha on the left panel in Figure 9, an ENVISAT observation is available on the 4-th day of the 35-days-repeat period when big off-peaks appear on the 25-th day of the same 35-days-repeat period. The right panel on Figure 9 displays the Serrinha station (red circle) with all ENVISAT observations available during the 25-th day (yellow circles). The inspection of contribution of all these observations to the analysis control variable at Serrinha (not shown here), we find that it is the observation number 4 that have
the highest impact on the analysis (and not the observation number 5, as it could be expected). This observation 4, located on a very small Negro tributary, has low discharge value and is responsible for the low corrected discharge at Serrinha after the assimilation step.

These abrupt variations are completely artificial and directly result from the assimilation processing. Indeed, for days with unrealistic peaks/off-peaks, there are multiple ENVISAT observations available on the basin, which impact many cells all over
the basin, even if they are located on other sub-catchments or tributaries. This is due to the construction of the error covariance matrices $[\mathbf{PH}^T]_{e,k}$ and $[\mathbf{HPH}^T]_{e,k}$. As these matrices are generated from the ensemble with a limited number of members, some spurious elements may appear in the matrices and link two cells that are very distant in the basin or even situated on different sub-basin. This first experiment highlights the necessity to treat the error covariance matrices to limit such spurious elements.

**4.2.2  SE1-diag results**

In the SE1-diag experiment, the error covariance matrices are forced to be diagonal. The objective of such processing on the error covariance matrices is to limit the impact of a given observation only to the observed cell. According to Table 4, the assimilation experiment allowed to reduce the global ENVISAT an in situ RMSE when compared to the free run. However, among all three experiments, it is the one which gives the worst global performances. In this experiment, the chaotic behaviour
of the mean analysis discharge is not present any more (not shown here). Nevertheless, the mean analysis discharge remains close to the free run discharge except for regular peaks/off-peaks at an observation time when it is closer to the observed discharge. Therefore, the information brought by only one local observation is not enough. With the localization (see next section), which results are presented in the following section, the information of several neighbouring VS is used and should more constrain the analysis discharge.

**4.2.3  SE1-local results**

SE1-local uses the localization treatment presented in section 3.2.4. Figure 10 displays the RMSEn evolution from the SE1-direct to the SE1-local experiment for both ENVISAT and gauge discharge. Green colours indicate that the SE1-local experiment reduced the RMSEn compared to the SE1-direct experiment while yellow to red colours indicate that the SE1-local





experiment increased them. The RMSEn is mostly improved over the entire basin and more particularly along major right-bank tributaries. However, the RMSEn are generally degraded along the mainstream. These maps show the good performances of the localization method over tributaries. Now, it appears that, compared to gauge discharges (see Figure 10b), the SE1-direct experiment gives better results, especially along the mainstream. Indeed, Table 5 details the local $\text{RMSEn}_i^{a,\dagger}$ at ENVISAT/insitu

stations located along the mainstream and confirms that SE1-direct gives better results. As the global RMSEn is defined as the mean of RMSEn weighted by the maximum discharge at the station (see Eq. 27), this explains why the SE1-direct experiment gives a better global RMSEn compared to gauge discharge.

Then, Figure 11 displays the mean analysis discharge for the SE1-local compared to the free run discharge, with corresponding ENVISAT discharge and gauge discharge, at the twelve in situ stations already used in Figures 5 and 8. Except for stations

along the mainstream (Figure 11,1-3) and also at Boa Sorte (Figure 11,12), the analysis discharge shows less sharp variations. From these results, we can say that the localization scheme is necessary and improves the assimilation.

### 4.2.4 Discussions on the localization scheme

The localization mask has been built to avoid the effect of spurious correlations between distant cells or ones situated on different sub-basins. The current localization scheme meets this constraint. Indeed, results for the SE1-local experiment are

globally improved compared to the previous experiments.

Nevertheless, along the mainstream, the initial experiment without localization gives better results. We can interpret that by the fact that discharge along the mainstream integrates hydrological processes from all the upstream basins. So when, in the SE1-local experiment, we limit the impact of the observation to only close cells, we suppress part of the information brought by distant cells to mainstream cells.

Therefore, the current localization mask should be improved. The main difficulty here is to determine the size of the influence area for each observation. Currently, this size is predetermined and is constant in time according to averaged flow velocity. A potential development is to consider an influence area size that can vary in time, according to the hydrological season (high-flow/low-flow season). For example, during high-flow season, the flow velocity is higher so is the size of the influence area. Thus, the error covariance matrices would depend on the river time and space dynamic (as if there were defined from a well-

sampled and significant ensemble).

### 4.3 Impact of the chosen control variables

In the second series of experiments, all of them uses the localization scheme (see section 3.2.4) to correct different types of state variables. After assimilating ENVISAT discharge to correct river initial storage (SE1-local experiment), we are testing in a second experiment the assimilation of ENVISAT discharge to correct river final storage (SE2-local) and, in a last experiment,

the assimilation of ENVISAT discharge to directly correct river discharge (SE3-direct). These two other experiments need to use an empirical relationship (see Eq. 3.1.1) linking simulated river final storage to simulated discharge. For the SE2-local experiment, the formula is used to convert analysis final rive storage to discharge. Indeed, experiment statistics are based on discharge and, when correcting the final river storage, we do not have an equivalent discharge. For the SE3-local experiment,



the formula is used during the assimilation steps to convert the analysis discharge into an equivalent river storage to propagate in time the corrected discharge.

Table 6 displays the global RMSEn for the three experiments compared to the free run global statistics. For all experiments, the assimilation enables to improve the RMSEn compared to the free run. Also, compared to both ENVISAT and gauge discharge, SE3-local experiment (discharge is the control variable) gives the best results, followed by SE1-local experiment (initial river storage is the control variable). Finally, it is SE2-local experiment (final river storage is the control variable) that gives the worst results, even if it is still improving the RMSEn compared to the free run.

Figures 12-13 display, for each ENVISAT (Figure 12a and 13a) and for each in situ stations (Figure 12b and 13b), mean RMSEn difference (in percent) between SE1-local experiment and SE2-local (Figure 12) and between SE1-local experiment and SE3-local experiment (Figure 13). Figure 12 shows a slight increase of the RMSEn in SE2-local experiment globally over the Amazon basin except for some basin heads. Also, the upstream part of the mainstream is more degraded (RMSEn increased of more than 60%). These degraded results imply that the assimilation of discharges may not be adapted to correct the final river stock. However, we need to keep in mind that the analysis discharges are determined from the analysis final river stock using Eq. 3.1.1. The bad SE2-local experiment results can either be due to bad assimilation results or to an unadapted formula to convert the final river storage into discharge. On the other hand, SE3-local experiment gives better general results. As in Table 6 and Figure 13, the SE3-local experiment shows a global improvement of the RMSEn compared to the SE1-local experiment (apart from a few cells upstream the Amazon mainstream). Indeed, even if Eq. 3.1.1 is still used to convert the analysis discharges back into river stock, it is used within the assimilation experiment (and not afterwards as for the SE2-local experiment). Therefore, the formula uncertainties are accounted for within the EnKF. Also, as the observed discharges are directly used to correct the simulated discharge, it appears logical that the assimilation gives better results as the link between the observed and the simulated variables is immediate.

## 5 Discussions

From the different approaches tested in this paper, it appears that there is not one specific configuration that gives the best results for all rivers, when compared to both ENVISAT and gauge discharges. On the contrary, the most effective configuration depends on the size and location of the rivers. Along the river mainstream (the Solimões and the Amazon in Figure 1a), the SE1-direct experiment clearly gives the best results (see the 3 first rows in Table 7). When the contribution of observations on tributaries is suppressed with the localization, the assimilation is less effective along the mainstream cells (see pannels 1 to 3 in Figure 8 for SE1-direct and compare to the same pannels in Figure 11 for SE1-local). This could be due to the fact that discharge along the mainstream is the results of hydrological processes from the entire drainage area. So, using all available observations helps the EnKF to correct the most efficiently discharge on the mainstream. However, it is different for cells along tributaries. As presented on Table 7, the localization method improve assimilation results for most cells along tributaries compared to the SE1-direct experiment. Along these cells, the localization allows to suppress the impact of observation from different sub basins, especially the ones that are not connected to these cells.



Nevertheless, among all experiments (see Table 3), the one producing the best results globally is SE3-local where the localization method is used to correct directly the discharge. Therefore, the SE3-local configuration is used for an 8 years experiment, from September 25$^{th}$ 2002 (first date with and ENVISAT observation on the study domain) to September 24$^{th}$ 2010 (last date with an ENVISAT observation). At the basin scale, RMSEn between model outputs and gauge discharges is

reduced by 27.11% (it decreases from 96.71% to 70.49%) and RMSEn between model outputs and ENVISAT discharges is reduced by 63.28% (it decreases from 75.10% to 27.58%). $\text{RMSEn}_{\text{global}}^{f,\text{situ}}$ is high, because a large fraction of in situ stations (25 out of 108) are situated along very small tributaries or at basin heads, where the local $\text{RMSEn}^{f,\text{situ}}$ is largely over 100%. These very high $\text{RMSEn}^{f,\text{situ}}$ have a huge impact on the global $\text{RMSEn}_{\text{global}}^{f,\text{situ}}$ (despite the weighting used to calculate $\text{RMSEn}_{\text{global}}^{f,\text{situ}}$). If the statistics are computed using only cells with a $\text{RMSEn}^{f,\text{situ}}$ below 100%, we find that the global $\text{RMSEn}_{\text{global}}^{f,\text{situ}}$ is reduced by

14.66% (it goes from 49.80% to 42.50%) and the $\text{RMSEn}_{\text{global}}^{f,\text{o}}$ is reduced by 50.21% (it goes from 51.74% to 25.76%). This shows the limitation of this assimilation scheme, as ISBA-CTRIP resolution (roughly 50 km by 50 km) does not well simulate basin heads (rivers are too small to be correctly represented in coarse grid).

Figure 14 displays, for the 12 in situ stations (see Figure 4 for their locations) already used in Figures 5 and 8, the mean analysis discharge over the whole experiment time period (red line), which is compared to the free run discharge at the station

(blue line), the ENVISAT discharge (green markers) and the gauge discharge (black markers). Overall, analysis discharge is quite close to observed discharges (ENVISAT and in situ).

However, despite the use of the localization, the analysis discharge keeps presenting a quite chaotic behaviour : more particularly at Sao Paulo de Olivenca (Figure 14, panel 1), Manacapuru (Figure 14, panel 2) and during high flow season along right-bank tributaries (Figure 14, panel 7 to 12). This shows the limit of assimilating 35-days repeat period ENVISAT obser-

vations. If no data is missing at a given VS, it means that there will be, at the most, 11 available observations during one year. Moreover, in a state estimation context, only the model output state is corrected and not the model parametrization or, in our set-up, forcings. Therefore, if the model is not constrained by direct or neighbouring observations, it naturally goes back to free run discharge. The performance of the assimilation, with respect to the daily in situ data, is therefore often limited by the low ENVISAT observations frequency. In future works, it will be interesting to study the assimilation of similar data with a

higher frequency, such as the JASON-2 altimeter data (which has a 10 days repeat period, but a coarser spatial sampling).

## 6   Conclusions and perspectives

This study presents, over the Amazon basin, the assimilation of a satellite-derived discharge product into a large-scale hydrological model to correct its state variables. The remotely-sensed discharge data is derived from the ENVISAT nadir altimeter and is assimilated into the ISBA-CTRIP model using an Ensemble Kalman Filter. Five experiments were carried out over the

year 2009. For all experiments, the assimilation were able to reduce the modelling errors compared to both observed and gauge discharges.

The first experiments tested different definition of the background error covariance matrices, where the influence of a given observation is either reduced to the only observed cell (SE1-diag), or limited to a few close cells on the hydrological network





(SE1-local), or not limited and can potentially impact the entire basin (SE1-direct). Results showed that the complete stochastic matrices gave the best results along the mainstream and the localization treatment appeared necessary along the tributaries. The need for the localization is explained by the spurious elements in the error covariance matrix due to the limited ensemble size and the methodology used to generate it.

The last tests compared the corrections of different state variables: the river initial storage (SE1-local), or the river final storage (SE2-local), or the river discharge (SE3-local). The main difficulty with this different type of variables is, on one hand, the relationship from the control to the observed variables (gathered in the observation operator) and, on the other hand, the reciprocal relationship to generate inputs for the next DA cycle. Results showed that correcting river discharge gives the best global results over the entire basin, as the link between the observed and the corrected variables is the most straightforward.

Therefore, the ultimate experiment (SE3-local-long) uses the SE3-local configuration over the whole ENVISAT observation period (from Sept 2002 to 2010) and confirms the possibility to use such low-resolution remotely-sensed data into a large-scale model.

These experiments offer several perspectives. First, the localization treatment could be improved by combining the three tested approaches according to the cell's position on the river: discharge correction for cells along the mainstream should be

impacted by all available observations, while correction for cells on tributaries should be impacted only by close observations along the same sub-catchment. Moreover, the size of the area of influence for a given observations could also vary in time according to the season (high flow/low flow). Then, even though the discharge is mainly sensitive to water inflow, the SA also showed that discharge is very sensitive to a particular parameter, the groundwater time constant, that controls the time delay taken by the groundwater to flow toward the river (Emery et al., 2016). Therefore, discharge observations could be assimilated

to correct this parameter in the CTRIP model.

A main limitation of assimilating ENVISAT data is its low repeat period (one observation every 35 days, at best). Indeed, corrected discharges often present strong sudden variations between unobserved and observed dates, as the model goes back to its free run when it is not constrained by an observation. However, there are other satellite altimetry missions with different repeat period. For example JASON-2 (10-days repeat period from June 2008 to October 2016), JASON-3 (10-days repeat

period, launched in January 2016 -) or Sentinel-3A (27-days repeat period, launched in February 2016). Also, the incoming SWOT (Surface Water and Ocean Topography, launch scheduled for 2021) wide-swath altimetry mission will also provide a remotely-sensed discharge product. SWOT will have a 21 days repeat period, with an almost global spatial coverage thanks to its two 50 km-swaths. All this data could be combined ENVISAT data (during the overlapping period) within the assimilation scheme to have a denser network of observation over the study domain. Or, their data and all available nadir altimetry data

since the early 1990's could be used to get a better estimate of discharge (similar to a reanalysis) over a multi decadal time frame (Tourian et al., 2017).

There are several perspectives to improve the DA results. On one hand, to get a more realistic ensemble, other parameters can be perturbed along with the initial condition and the precipitation so that their uncertainties are also taken into consideration, like groundwater time constant or surface runoff. On the other hand, to improve corrected discharge, a smoothing data

assimilation technique could be used to remove the noisy aspect of the analysis discharge, like the Ensemble Kalman Smoother

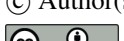



(Evensen and Leeuwen, 2000). Finally, the assimilation scheme presented in this study could be applied to other river basin in the world, as ISBA-CTRIP is a global LSM. However, more work is needed to apply the DA platform at global scale.

## Appendix A: Equations to compute river storage from discharge using the Manning formula

This appendix provides more details and the approximation used to derive Eq. 3.1.1. This equation allows converting simulated
discharges $Q_{i,k}^a$ to equivalent final river storage $S_{\mathrm{end},k,i}^a$ using the Manning formula. We chose to invert Eq. 8. Assuming that the discharge estimated by ISBA-CTRIP $Q_{i,k}^a$ is the instantaneous flow at the final time of the integration window:

$$Q_{k,i,[\mathrm{kg.s}^{-1}]}^a = L^{-1}v S_{\mathrm{end},k,i,[\mathrm{kg}]}^a \Longleftrightarrow Q_{k,[\mathrm{m}^3.\mathrm{s}^{-1}]}^a = \rho^{-1}L^{-1}v S_{\mathrm{end},k,i,[\mathrm{kg}]}^a.$$

To ease the notations, we will skip the units in the following equations knowing that discharges are expressed in $\mathrm{m}^3.\mathrm{s}^{-1}$ and water stock in kg. Also, we skip the $\kappa$ constant in Eq. 5 as it is equal to 1. Then, $\forall\, k,\, \forall\, i$:

$Q_{k,i}^a = L^{-1}\rho^{-1}s^{1/2}n^{-1}\left(\dfrac{Wh_S}{W+2h-S}\right)^{2/3} S_{\mathrm{end},k,i}^a,$

We suppose : $W >> h_S$

$Q_{k,i}^a \approx L^{-1}\rho^{-1}s^{1/2}n^{-1}h^{2/3}S_{\mathrm{end},k,i}^a,$

Yet $S = \rho L W h_S$, so

$Q_{k,i}^a \approx L^{-5/3}\rho^{-5/3}W^{2/3}s^{1/2}n^{-1}\left(S_{\mathrm{end},k,i}^a\right)^{5/3},$

15                                                                      (A1)

Finally giving Eq. 3.1.1:

$$S_{\mathrm{end},k,i}^a \approx \rho L W^{2/5}s^{-3/10}n^{3/5}\left(Q_{k,i}^a\right)^{3/5} \tag{A2}$$

with $\rho$ [$\mathrm{m}^3.\mathrm{kg}^{-1}$] the water density, $L$ [m] the river section length, $W$ [m] the river width, $s$ [-] the riverbed slope and $n$ [$\mathrm{s.m}^{-1/3}$] the Manning coefficient in the riverbed. Then, for experiments with discharges as control variables, the formula in
Eq. 3.1.1 will be used to turn back corrected discharges into river stock and propagate the model uo to the next observation time.

## Appendix B: Definition of error covariance matrices

The background error covariance matrices $[\mathbf{PH}^T]_{e,k}$ et $[\mathbf{HPH}^T]_{e,k}$ are estimated from the definition suggested by Evensen (2004); Moradkhani et al. (2005); Durand et al. (2008) such that:

$[\mathbf{PH}^T]_{e,k} = (n_e-1)^{-1}\left(\mathbf{X}_{e,k}^b - \overline{\mathbf{X}_{\bullet,k}^b}.\mathbb{1}_{n_e}^T\right)\left(\mathcal{H}(\mathbf{X}_{e,k}^b) - \overline{\mathcal{H}(\mathbf{X}_{\bullet,k}^b)}.\mathbb{1}_{n_e}^T\right)^T,$            (B1)





and

$$[\mathbf{HPH}^T]_{e,k} = (n_e - 1)^{-1} \left( \mathcal{H}(\mathbf{X}_{e,k}^b) - \overline{\mathcal{H}(\mathbf{X}_{\bullet,k}^b)}.\mathbb{1}_{n_e}^T \right) \left( \mathcal{H}(\mathbf{X}_{e,k}^b) - \overline{\mathcal{H}(\mathbf{X}_{\bullet,k}^b)}.\mathbb{1}_{n_e}^T \right)^T, \tag{B2}$$

with $\mathbf{X}_{e,k}^b$ the control matrix containing the $n_e$ control vector $\mathbf{x}_k^{b,[l]}$, $l = 1 \ldots n_e$ from the background ensemble such that

$$\mathbf{X}_{e,k}^b = \left[ \mathbf{x}_k^{b,[1]} \ \ldots \ \mathbf{x}_k^{b,[N_e]} \right]$$

and $\mathcal{H}(\mathbf{X}_{e,k}^b)$ the same control matrix expressed in the observation space such that

$$\mathcal{H}(\mathbf{X}_{e,k}^b) = \left[ \mathcal{H}(\mathbf{x}_k^{b,[1]}) \ \ldots \ \mathcal{H}(\mathbf{x}_k^{b,[n_e]}) \right].$$

$\overline{\mathbf{X}_{\bullet,k}^b}$ and $\overline{\mathcal{H}(\mathbf{X}_{\bullet,k}^b)}$ are the ensemble sample expectations of the control matrix $\mathbf{X}_{e,k}^b$ and its mapping on the observation space $\mathcal{H}(\mathbf{X}_{e,k}^b)$ respectively such that

$$\overline{\mathbf{X}_{\bullet,k}^b} = \frac{1}{n_e} \sum_{l=1}^{n_e} \mathbf{x}_k^{b,[l]} \quad \overline{\mathcal{H}(\mathbf{X}_{\bullet,k}^b)} = \frac{1}{n_e} \sum_{l=1}^{n_e} \mathcal{H}(\mathbf{x}_k^{b,[l]}).$$

The vectors dimension are $n_x$ and $n_{y,k}$ respectively and $\mathbb{1}_{n_e}$ is a vector of size $n_e$ containing only 1s.

## Appendix C: Perturbations of precipitations

The ensemble of perturbed precipitation fields $\widetilde{\mathbf{F}}_e$ is defined such that:

$$\widetilde{\mathbf{F}}_e = \left\{ \widetilde{\mathbf{F}}^{[1]}, \quad \widetilde{\mathbf{F}}^{[2]}, \quad \ldots \quad \widetilde{\mathbf{F}}^{[n_e]} \right\} = \left\{ \varphi_p^{[1]}.\mathbf{F}, \quad \varphi_p^{[2]}.\mathbf{F}, \quad \ldots \quad \varphi_p^{[n_e]}.\mathbf{F} \right\}, \tag{C1}$$

where:

- $\mathbf{F}$ is the two-dimensional field of precipitation forcing before perturbation (with a time-step of 3 hours),

- $\widetilde{\mathbf{F}}^{[l]}$, for $l = 1 \ldots n_e$, is the $l$-th perturbed precipitation field,

- $\varphi_p^{[l]}$, for $l = 1 \ldots n_e$, is the $l$-th multiplying uniformly-distributed field of $\mathbf{F}$ to generate $\widetilde{\mathbf{F}}^{[l]}$.

The precipitation field $\mathbf{F}$ is then perturbed by applying a random multiplying field such that

$$\varphi_p^{[l]} = (1 - \eta^{\mathbf{F},[l]}) + 2\mathbf{U}_{\mathbf{F}}^{[l]}.\eta^{\mathbf{F},[l]}, \tag{C2}$$

where

- $\mathbf{U}_{\mathbf{F}}^{[l]}$ is random field following a uniform law between 0 and 1,

- $\eta^{\mathbf{F},[l]}$ is a scalar representing the relative error of the precipitations.





Therefore, $\varphi_p^{[l]}$ is a random field following a uniform law between $(1 - \eta^{\mathbf{F},[l]})$ and $(1 + \eta^{\mathbf{F},[l]})$.

The precipitation relative error $\eta^{\mathbf{F}}$ quantifies the uncertainties into the precipitation intensity. The variable $\eta^{\mathbf{F}}$ is different for each member of the ensemble and follows a Gaussian law with expectation $\overline{\eta^{\mathbf{F}}} = 30\%$ and standard deviation $\sigma_{\eta^{\mathbf{F}}} = 0,1\%$ (Clark et al., 2008; Paiva et al., 2013b).

The fields $\mathbf{U}_{\mathbf{F}}^{[l]}$, for $l = 1 \ldots n_e$, allow to introduce a time and space correlation in the precipitation error and are generated with the algorithm presented in Evensen (2003). This algorithm generates two-dimensional Gaussian random fields $\mathbf{S}^{[l]}$ with a zero-mean and a space-correlation length of $\mathrm{e}^{-1}$. These Gaussian random fields are turned into uniform random fields by applying the complementary error function erfc():

$$\mathbf{U}_{\mathbf{F},k'}^{[l]} = \frac{1}{2}\mathrm{erfc}\left(\frac{\mathbf{S}_{k'}^{[l]}}{\sqrt{2}}\right), \tag{C3}$$

where $k'$ is the atmospheric forcing proper time-step, equal to 3 hours in ISBA-CTRIP, and shorter than the ISBA-CTRIP outputs time-step, equal to 24 hours. The space-PDF of $\mathbf{U}_{\mathbf{F}}^{[l]}$ decreases of $\mathrm{e}^{-1}$ when the distance is equal to the space-correlation length $\tau_x$ (here, the $x$ letter exceptionally denotes the spacial dimension). For the simulations, $\tau_x$ is fixed to $1,0°$ (Clark et al., 2008; Paiva et al., 2013b) and is invariant from one member to another and from one assimilation cycle to another.

For the time correlation, the parameter $\vartheta^{[l]}$

$$\vartheta^{[l]} = 1 - \frac{\Delta k'}{\tau_k^{[l]}} \tag{C4}$$

determines the time correlation length between the different fields $\mathbf{S}_{k'}^{[l]}$. It is concretely generated by combining the random field from the previous time step $\mathbf{S}_{k'-1}^{[l]}$ and an auxiliary random field $\mathbf{W}_{k'}^{[l]}$ with the same properties such that:

$$\mathbf{S}_{k'}^{[l]} = \vartheta^{[l]}\mathbf{S}_{k'-1}^{[l]} + \sqrt{1 - (\vartheta^{[l]})^2}\mathbf{W}_{k'}^{[l]}, \tag{C5}$$

with $\Delta k' = 3$ hours is the forcing time step and $\tau_k^{[l]}$ the time constant characterizing $\vartheta^{[l]}$. $\vartheta^{[l]} = 0$ generates a white noise
(which means a perturbation uncorrelated in time) while $\vartheta^{[l]} = 1$ makes the perturbation constant in time. The variable $\tau_k$ takes a different value for each member as it follows a gaussian law with an expectation equal to $\overline{\tau_k} = 12$ hours (or 43200 seconds) and a standard deviation $\sigma_{\tau_k} = 3$ hours (or 10800 seconds). These values are chosen so that the time correlation has effects during assimilation window of one day. All variables used to generate the ensembles with their value are summarized in the Table A1.

*Competing interests.* No competing interests are present

*Acknowledgements.* This work has been performed using HPC resources from CALMIP (Grant 2016-P1408). The GSWP3 team is acknowledged for letting the authors use their different forcing fields. This work was supported by the CNES, through a grant from Terre-Océan-Surfaces Continentales-Atmosphére (TOSCA) committee attributed to the project entitled "Towards an improved understanding of the global





hydrological cycle using SWOT measurements". The European Space Agency (ESA) is also thanked for providing to the scientific community observations from the RA2 altimeter embarked on ENVISAT. C.M. Emery was supported by a CNES/région Midi-Pyrénées PhD grant.





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





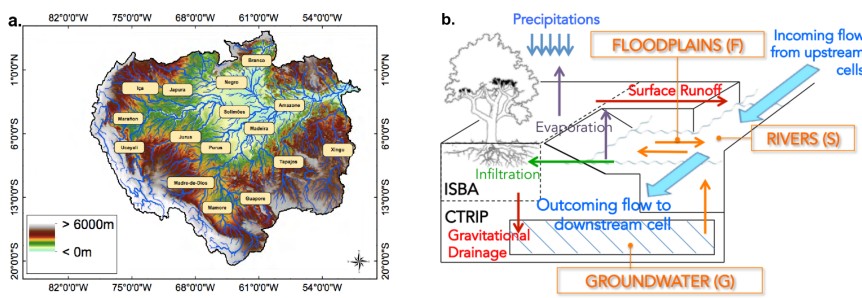

**Figure 1.** (a) The Amazon basin main tributaries and rivers with the underlying topography from SRTM. (b) Schematic representation of ISBA-CTRIP system for a given grid cell. ISBA surface runoff ($Q_{\mathrm{ISBA,sur}}$) flows into the river/surface reservoir $S$, ISBA gravitational drainage ($Q_{\mathrm{ISBA,sub}}$) feeds groundwater reservoir $G$. The surface water is transferred from one cell to another following the TRIP river routing network.



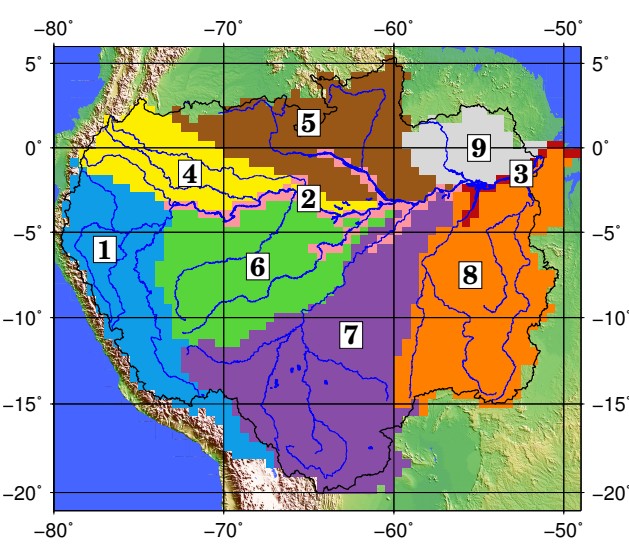

**Figure 2.** Map of hydro-geomorphological zones defined over the Amazon basin.





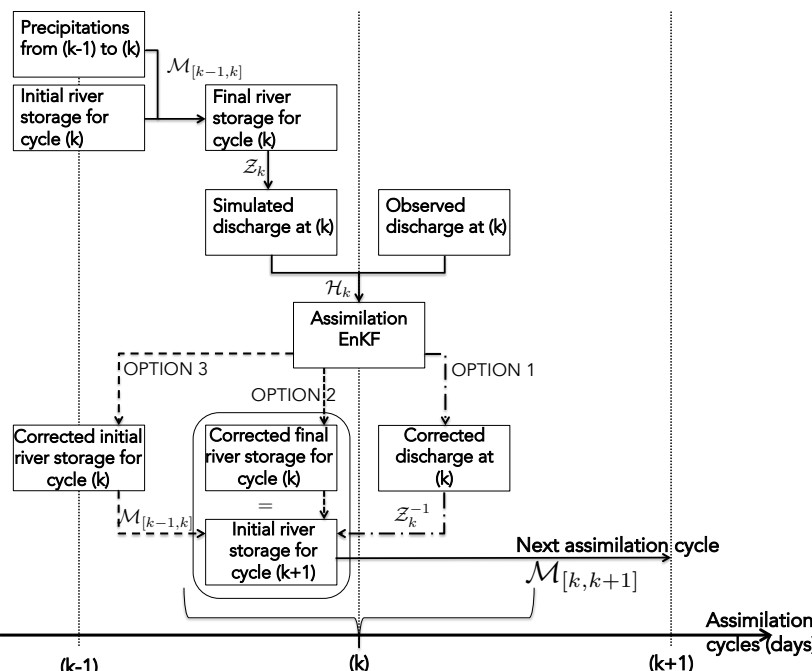

**Figure 3.** General framework of the DA method at a $k$-th assimilation cycle. Figure reads from top to bottom and from left to right. The three main variables involved are the river initial storage, the river final storage and the river discharge. $\mathcal{M}_{[k-1,k]}$ is the model operator that maps the initial river storage into final river storage, $\mathcal{Z}_k$ is the diagnostic operator and $\mathcal{H}_k$ is the observation operator that maps simulated discharge into observed discharge for assimilation.



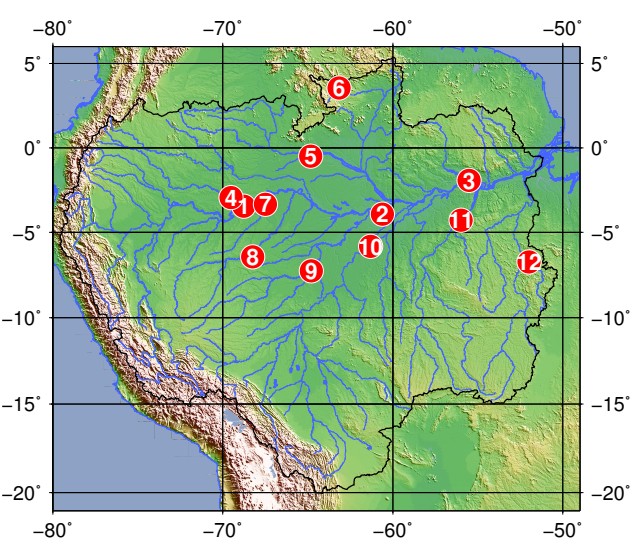

**Figure 4.** Map of the twelve in situ stations used to evaluate assimilation performance: 1. São Paulo de Olivenca (Solimões), 2. Manacapuru (Solimões), 3. Óbidos (Amazonas), 4. Ipiranga (Putumayo/Icá), 5. Serrinha (Negro), 6. Uaicás (Branco), 7. Porto Seguro (Jutaí), 8. Santos Dumont (Juruá) 9. Lábrea (Purus), 10. Manicoré (Madeira), 11. Itaituba (Tapajós), 12. Boa Sorte (Xingu).





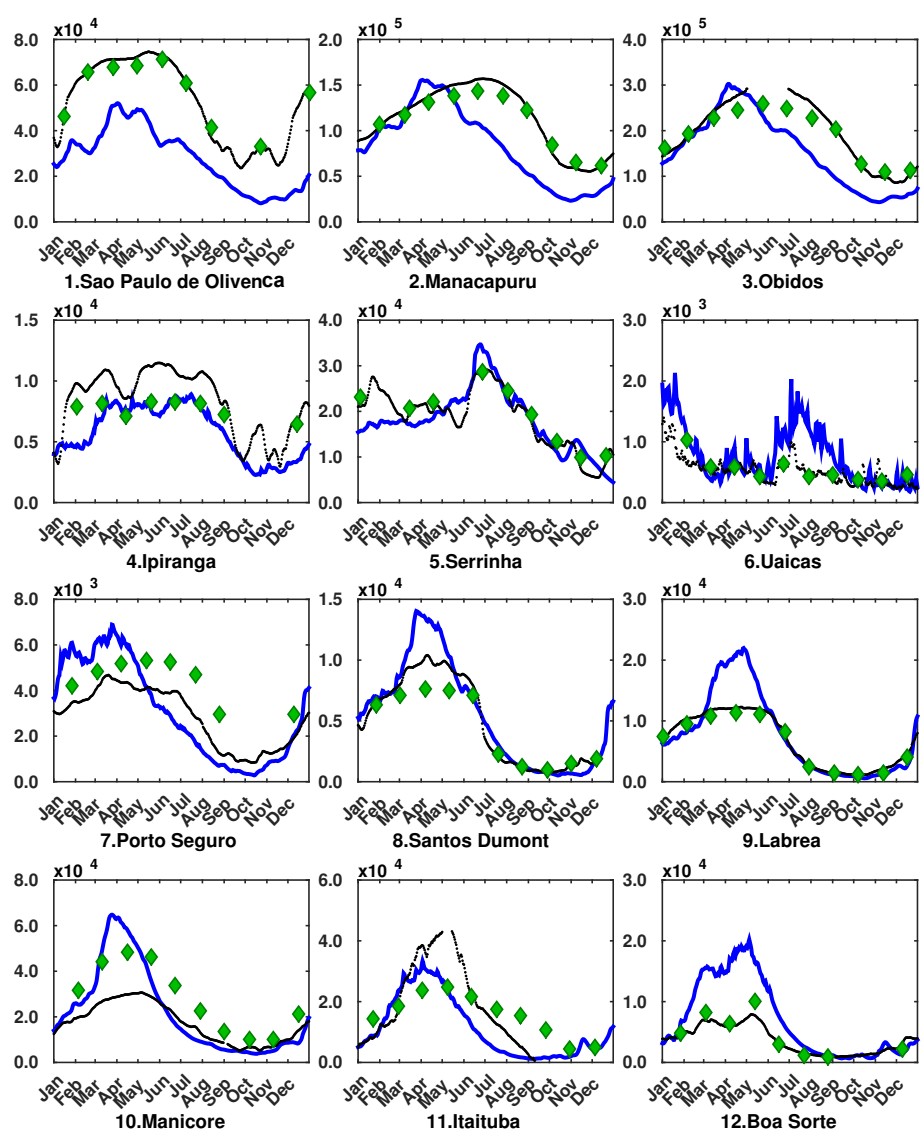

**Figure 5.** Comparison between ISBA-CTRIP free run (blue line), ENVISAT-derived observed discharges (green markers) and ANA gauge discharges (black dots) over the year 2009. For each panel, the x-axis represents time (in days) and the y-axis represents discharge (in $m^3.s^{-1}$).



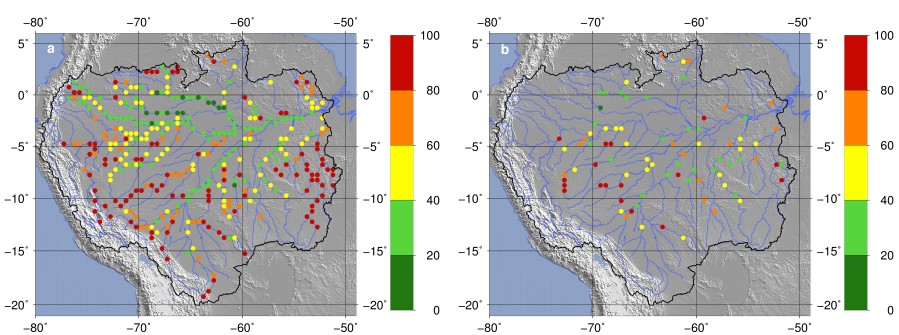

**Figure 6.** RMSEn for the free run simulation compared to the ENVISAT discharges (a) and the gauge discharges (b).



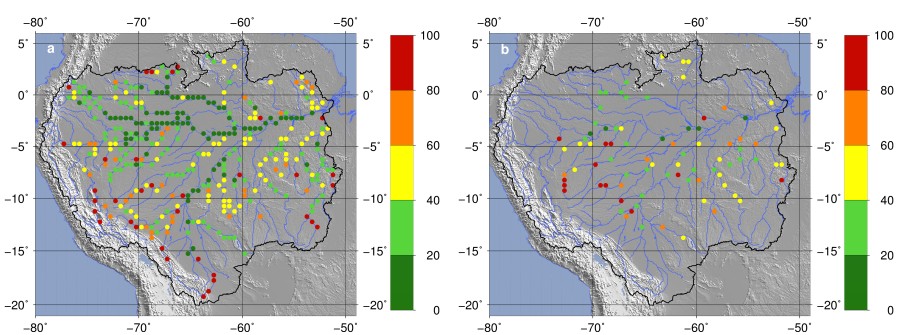

**Figure 7.** Analysis RMSEn for the SE1-direct experiment with respect to (a) the ENVISAT discharge and (b) gauge discharge.





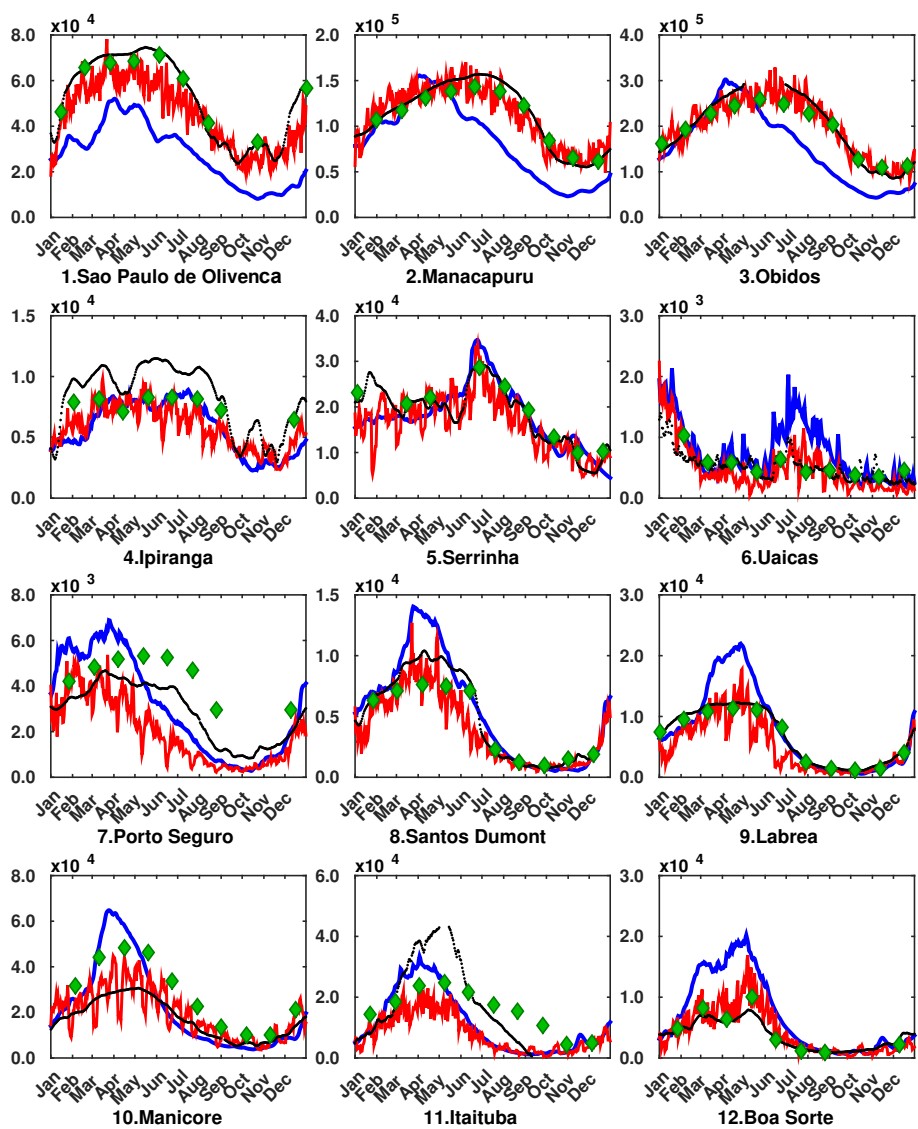

**Figure 8.** SE1-direct ensemble mean analysis discharge (red line) compared to the free run discharge (blue line), the ENVISAT observed discharges (green markers) and the measured gauge discharges (black dots) over the year 2009. For each panel, the x-axis represents time (in days) and the y-axis represents discharge (in $m^3.s^{-1}$).



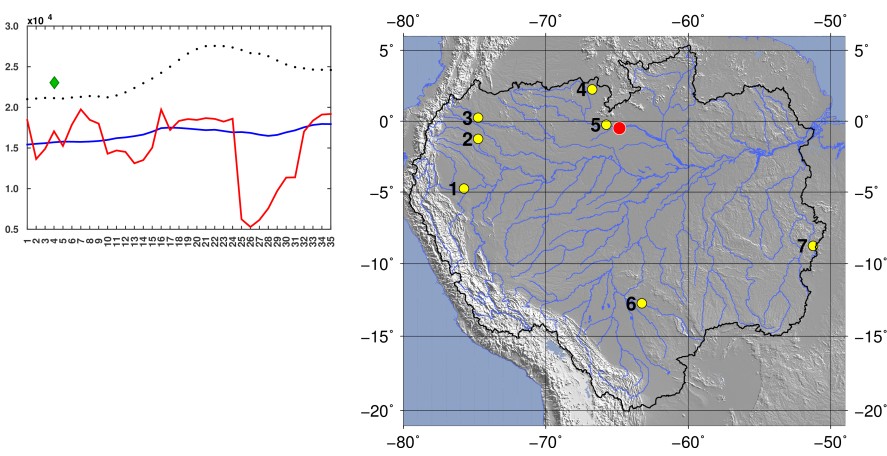

**Figure 9.** (Left) Same as Figure 8, panel 5 but only over the 35 first day of simulation. (Right) Location of all active ENVISAT VS on the 25-th day of the assimilation (yellow circles) compared to the location of the Serrinha stations (red circle).




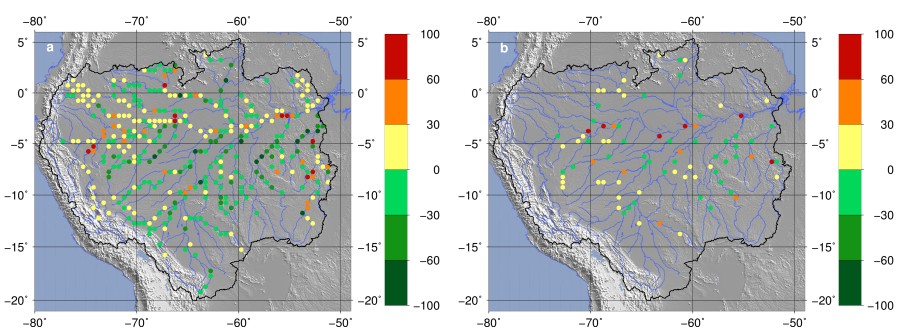

**Figure 10.** Analysis RMSEn difference between SE1-direct and the SE1-local experiment with respect to (a) the ENVISAT discharge and (b) gauge discharge. Negative RMSEn differences (green colours) mean that the results of the SE1-local experiment are better than the SE1-direct results at the given CTRIP cell. Positive RMSEn differences (yellow, orange and red colours) mean that the results of the SE1-direct experiment are better that the SE1-local results at the given CTRIP.



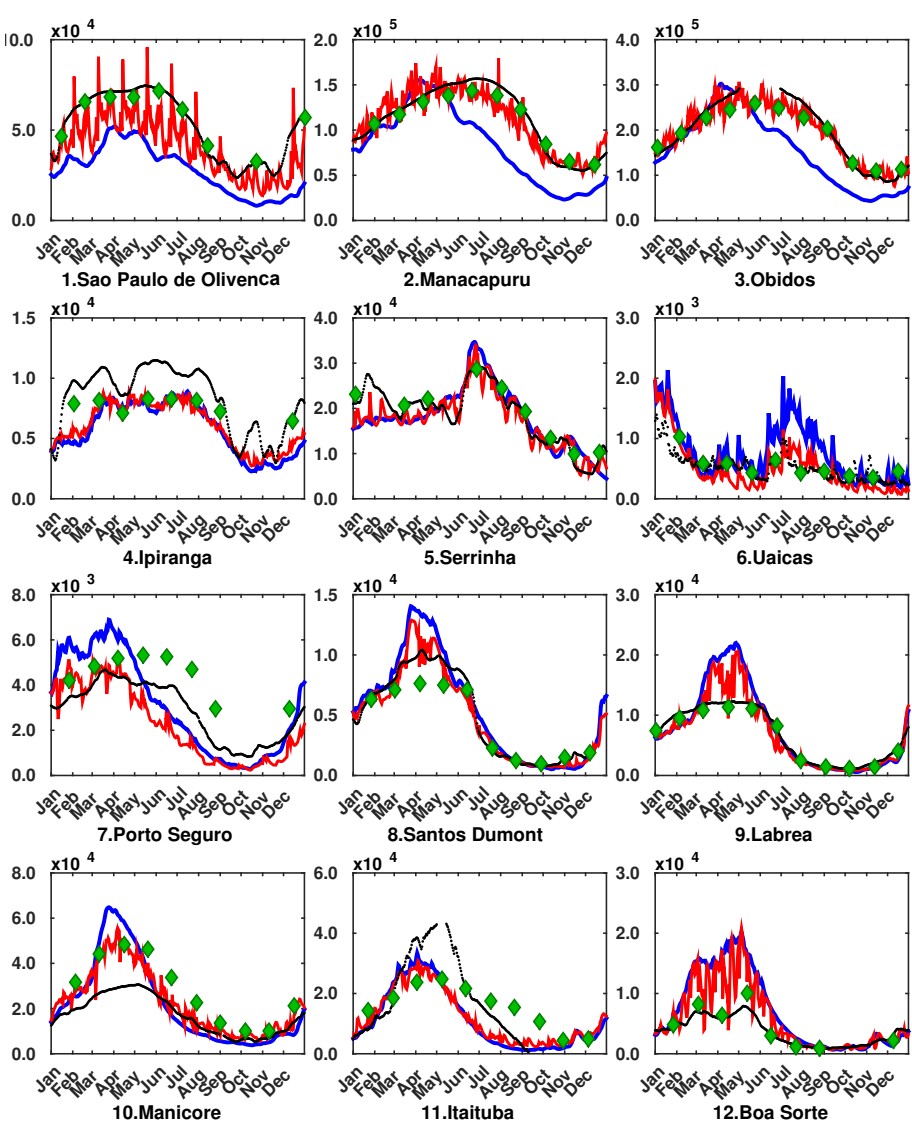

**Figure 11.** SE1-local ensemble mean analysis discharge (red line) compared to the free run discharge (blue line), the ENVISAT observed discharges (green markers) and the measured gauge discharges (black dots) over the year 2009. For each panel, the x-axis represents time (in days) and the y-axis represents discharge (in $m^3.s^{-1}$).



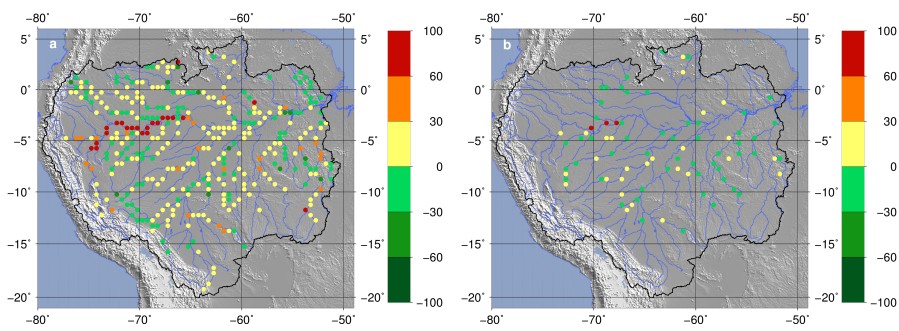

**Figure 12.** Analysis RMSEn differences between SE1-local and SE2-local experiments with respect to (a) the ENVISAT discharge and (b) gauge discharge.





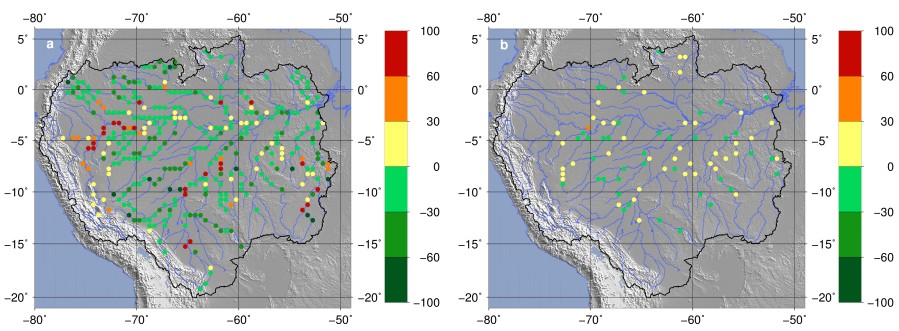

**Figure 13.** Analysis RMSEn differences between SE1-local and SE3-local experiments with respect to (a) the ENVISAT discharge and (b) gauge discharge.



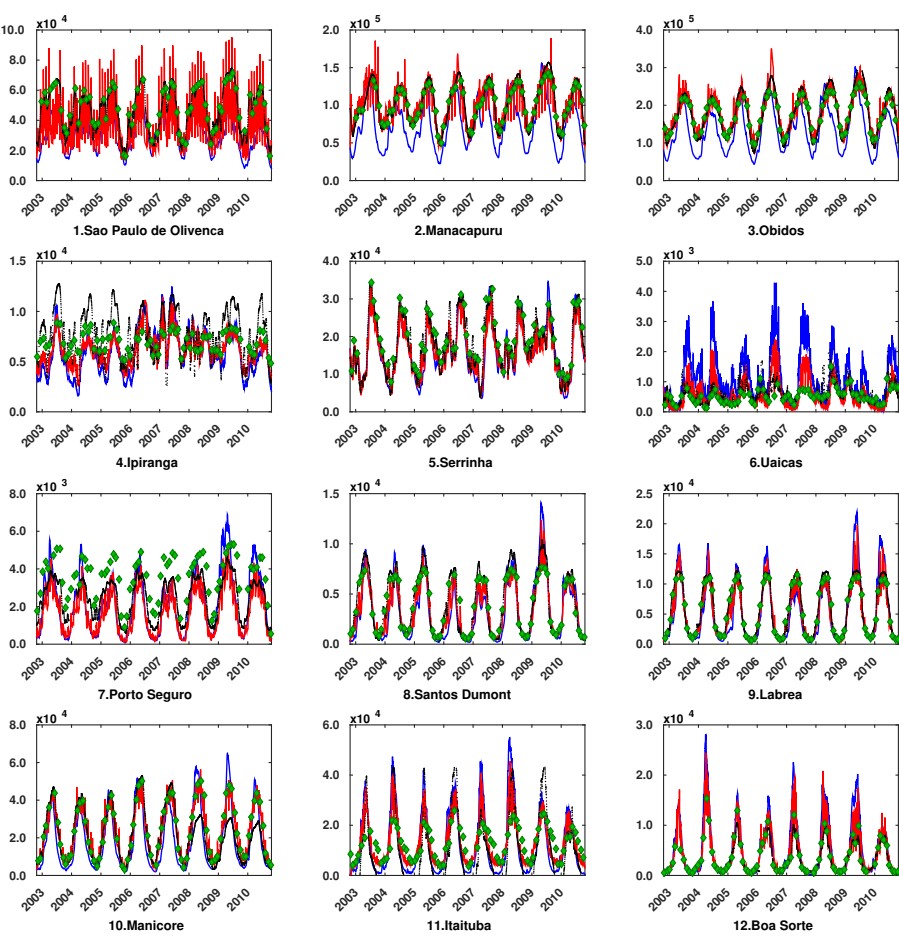

**Figure 14.** SE3-local ensemble mean analysis discharge (red line) compared to the free run discharge (blue line), the ENVISAT observed discharges (green markers) and the measured gauge discharges (black dots) from September, 25$^{th}$ 2002 et for 8 years. In each panel, the x-axis represents time (in days) and the y-axis represents discharge (in m$^3$.s$^{-1}$).





**Table 1.** Constant values used to generate the background control ensemble $\mathbf{X}^b_{e,k}$, the observation ensemble $\mathbf{Y}^o_{e,k}$ and the model observation ensemble $\mathcal{H}(\mathbf{X}^b_{e,k})$. $k$ is the assimilation index and $s$ is the basin zone index.

| Variable | Description | Value | Eq. |
|:---:|:---:|:---:|:---:|
| $n_e$ | Ens size | 101 | - |
| $n_x$ | Control space size | 2028 | - |
| $n_y$ | Obs space size | $\in [0, 15]$ | - |
| $\eta^o$ | Obs error | $0,2$ | 14 |
| | Error | $s = 2, 3, k = 1 : 0,25$ | |
| | on | $s = 2, 3, k > 1 : 0,05$ | |
| $\sigma_{\eta^x_{s,k}}$ | initial | $s = 1, 4 : 9, k = 1 : 0,10$ | 23 |
| | condition | $s = 1, 4 : 9, k > 1 : 0,02$ | |





**Table 2.** Size of the influence area for the localization process.

| Zone | Real Max Dist [km] | Rounded Max Dist [km] | Influence Area [number of cells] |
|:---:|:---:|:---:|:---:|
| 1 | 84.16 | 100 | 2 |
| 2 | 174.24 | 200 | 4 |
| 3 | 233.80 | 250 | 5 |
| 4 | 93.29 | 100 | 2 |
| 5 | 82.03 | 100 | 2 |
| 6 | 80.03 | 100 | 2 |
| 7 | 69.57 | 100 | 2 |
| 8 | 87.86 | 100 | 2 |
| 9 | 67.02 | 100 | 2 |



**Table 3.** Presentation of the different state estimation experiments. The "SE" acronym stands for "State Estimation", the index "1", "2" or "3" are to differentiate the control variables ("1"=initial river storage, "2"=final river storage and "3"=discharge) and the suffix "direct", "diag" and "local" indicates the localization scheme ("direct"=without localization, "diag"=diagonal error covariance matrices and "local"=with localization).

| Exp. Name | Control Variable | Localization scheme |
|:---:|:---:|:---:|
| SE1-direct | initial storage | No - $[\mathbf{PH}^T]_{e,k}$ and $[\mathbf{HPH}^T]_{e,k}$ defined in Eq. B1-B2 |
| SE1-diag | initial storage | No - Diagonal $[\mathbf{PH}^T]_{e,k}$ and $[\mathbf{HPH}^T]_{e,k}$ |
| SE1-local | initial storage | Yes - see Section 3.2.4 |
| SE2-local | final storage | Yes - see Section 3.2.4 |
| SE3-local | discharge | Yes - see Section 3.2.4 |





**Table 4.** Global statistics for experiments with different the localization scheme.

| Statistics | Free run | SE1-direct | SE1-diag | SE1-local | Units |
|---|---|---|---|---|---|
| | $(* = f)$ | $(* = a)$ | $(* = a)$ | $(* = a)$ | |
| $\mathbf{RMSE}^{*,\mathbf{o}}_{\mathbf{global}}$ | $8.3795 \times 10^3$ | $3.2110 \times 10^3$ | $4.8626 \times 10^3$ | $2.4377 \times 10^3$ | $\mathrm{m}^3.\mathrm{s}^{-1}$ |
| $\mathbf{RMSEn}^{*,\mathbf{o}}_{\mathbf{global}}$ | 71.12 | 36.30 | 44.30 | 31.16 | % |
| $\mathbf{RMSE}^{*,\mathbf{situ}}_{\mathbf{global}}$ | $7.1478 \times 10^3$ | $4.2489 \times 10^3$ | $5.4300 \times 10^3$ | $4.1542 \times 10^3$ | $\mathrm{m}^3.\mathrm{s}^{-1}$ |
| $\mathbf{RMSEn}^{*,\mathbf{situ}}_{\mathbf{global}}$ | 68.96 | 57.54 | 63.12 | 58.44 | % |





**Table 5.** Local $\overline{\text{RMSEn}_i^{a,situ}}$ for the SE1-direct and SE1-local experiments at in situ stations along the mainstream (from the most upstream to the most downstream).

| Station | $\overline{\text{RMSEn}_i^{a,situ}}$ | |
| --- | --- | --- |
| | SE1-direct | SE1-local |
| Tamishiyacu | 100.97 | 92.54 |
| Tabatinga | 22.24 | 38.81 |
| São Paulo de Olivenca | 21.77 | 34.86 |
| Itapéua | 17.20 | 28.58 |
| Manacapuru | 18.15 | 30.38 |
| Jatuarana | 19.72 | 31.01 |
| Óbidos | 16.60 | 28.11 |





**Table 6.** Global statistics for experiments with different types of control variables.

| Statistics | Free run ($* = f$) | SE1-local ($* = a$) | SE2-local ($* = a$) | SE3-local ($* = a$) | Units |
|---|---|---|---|---|---|
| $\mathbf{RMSE}^{*,o}_{global}$ | $8.3795 \times 10^3$ | $2.4377 \times 10^3$ | $4.3069 \times 10^3$ | $2.2298 \times 10^3$ | $m^3.s^{-1}$ |
| $\mathbf{RMSEn}^{*,o}_{global}$ | 71.12 | 31.16 | 36.37 | 24.73 | % |
| $\mathbf{RMSE}^{*,situ}_{global}$ | $7.1478 \times 10^3$ | $4.2489 \times 10^3$ | $5.4300 \times 10^3$ | $4.1542 \times 10^3$ | $m^3.s^{-1}$ |
| $\mathbf{RMSEn}^{*,situ}_{global}$ | 68.96 | 58.44 | 61.69 | 54.46 | % |





**Table 7.** Statistics between analysis and in situ stations for the different assimilation experiments

| Statistics | $\overline{\text{RMSEn}_i^{*,\text{situ}}}$ | | | |
| --- | --- | --- | --- | --- |
| | Free run | SE1-direct | SE1-local | SE3-local |
| **1. São Paulo de Olivenca** | 49.67 | *21.77* | 34.86 | 40.98 |
| **2. Manacapuru** | 36.01 | *18.15* | 30.38 | 30.76 |
| **3. Óbidos** | 34.65 | *16.60* | 28.11 | 28.33 |
| **4. Ipiranga** | 34.63 | 35.43 | *32.75* | 33.54 |
| **5. Serrinha** | *20.65* | 26.82 | 23.99 | 28.24 |
| **6. Uaicás** | 79.53 | 51.28 | 51.33 | *48.92* |
| **7. Porto Seguro** | 44.16 | 46.32 | 39.81 | *38.93* |
| **8. Santos Dumont** | 28.37 | 35.54 | *27.55* | 33.63 |
| **9. Lábrea** | 50.62 | 40.39 | 40.33 | *39.86* |
| **10. Manicoré** | 72.36 | *35.04* | 54.38 | 61.96 |
| **11. Itaituba** | *43.33* | 66.43 | 47.50 | 46.23 |
| **12. Boa Sorte** | 149.38 | *58.99* | 112.58 | 82.75 |





**Table A1.** Constant values used to perturb the precipitation fields. $k$ is the assimilation index.

| Variable | Description | Value | Eq. |
|:---:|:---:|:---:|:---:|
| $\tau_x$ [∘] | Precip spatial corr. | 1.0 | |
| $\overline{\eta^{\mathbf{F}}}$ | Precip relative error mean | $0,3$ | C2 |
| $\sigma_{\eta^{\mathbf{F}}}$ | Precip relative error mean | $0,1$ | C2 |
| $\overline{\tau_k}$ [s] | Precip temp corr mean | 43200 | C4 |
| $\sigma_{\tau_k}$ [s] | Precip temp corr std | 10800 | C4 |