# Peer review of "Large scale hydrological model river storage and discharge correction using satellite altimetry-based discharge product"

_Hydrology and Earth System Sciences, 2017_

## Referee Comment (RC1) · R. Hut (Referee) · 3 Nov 2017

This review was jointly done by Rolf Hut and Zhenwu Wang.

This study applies Ensemble Kalman filter with two localization methods to assimilate discharge data into a relatively large scale hydrological model. As far as I can tell, this paper should the first case study in hydrology field to apply an EnKF with localization in such a high dimensional model. The academic value of this research is great and their exploration is valuable to the readership of HESS. After reviewing the entire paper, I have some concerns that need to be clarified and explained. Besides that, I have some minor suggestion and tips. 1. In the introduction, the third paragraph is about the

resources of model errors. In my opinion, a description of the accumulation of model errors and other uncertainties in model's prediction which can lead to the collapse of the model should be added. I think this part can be regarded as a part of the explanation of the necessity of data assimilation. I recommend to shorten the section on model error sources and add some sentences on the impact of accumulation of uncertainties. 2. In Lopez Lopez (2016) discharge data was assimilated into a model for the Rhine basin. I think this paper should be added to the introduction. (López López, P., Wanders, N., Schellekens, J., Renzullo, L. J., Sutanudjaja, E. H., and Bierkens, M. F. P.: Improved large-scale hydrological modelling through the assimilation of streamflow and downscaled satellite soil moisture observations, Hydrol. Earth Syst. Sci., 20, 3059-3076, https://doi.org/10.5194/hess-20-3059-2016, 2016) 3. Page 3, in last second paragraph, the aim of this study only mentioned that EnKF is applied and also said something about the model and observations. From my point of view, the using of localization methods should be mentioned. The localization methods are the crucial key to this case study and without localization, the academic value of this research will mostly be as a case study of an otherwise known method into a new geographical area. 4. In section 2, I think the description of the CTRIP RRM is excessive. It would be better to make the context of this model shorter and simpler. Maybe the authors can reference one of their earlier papers on the model and point to that for the model description. 5. In section 3, it has the similar issue just like section 2, the explenation of the fairly standard EnKF is too long. 6. Section 3.2.4 (Localization) does not belong to "3.2 Generating the ensembles". In my opinion, the following structure of section 3 is better. Firstly, introduce the control variables and observations separately. And then, give a short introduction to EnKF theory. Following, present how to implement the EnKF with localization specifically. Last part includes diagnostics and experiment set up.

7. In page 14, section 3.2.4, the second paragraph, it said that there are three localization methods. I prefer to state that there are two common localization methods, namely local analysis (R localization ) and covariance localization (B localization) . These two

methods can be found in following two papers. -Balance and Ensemble Kalman Filter Localization Techniques (doi:10.1175/2010MWR3328.1) -Relation between two common localisation methods for the EnKF (doi: 10.1007/s10596-010-9202-6) . 8. The names of the different localization schemes should align with the common names in the data assimilation field. I recommend the terms in those two papers in point 7, above. In table 3 and corresponding parts in the main body of this research should change "-local suffix" to "Local analysis" or "R localization" and also replace "-diagonal suffix" with "covariance localization" or "B localization". 9. On page 16, the part before 3.3, it describes how to get the localization matrix. In this study, the author used localization matrix to multiply covariance matrix directly. This way is not wrong but it differs from the most common way to implement the localization methods. Can you use equations to display the formulation of localization matrix? This is helpful for readers to understand your localization methods. 10. In table 1, the size of the ensemble is 101. The authors do not justify the choice of exactly 101 ensemble members in the paper. It is not possible for the reviewer to see if the ensemble size represent the distribution of model states properly? Could the authors use some figures or the rank histogram of the ensemble to show the gaussianity of the ensemble? Otherwise, could the authors justify the choice for 101 ensemble members? 11. In data assimilation applications with localization usually a common localization function is used. Common example is a fifth order function of Gaspari and. I didn't find the description of the localization function in this paper. If I missed it, please point out its location. If the authors didn't use it, could the authors explain reasons and considerations? 12. It is common that localization methods can cause imbalance. The analysis of imbalance can show the performance of localization method in specific application. I recommend adding the imbalance analysis. If the authors think it is unnecessary, could the authors explain the reasons? 13. I am a bit confused about the chosen localization scales. The "diagonal error covariance matrix" in this paper is to apply B localization method to form the localized covariance matrix. In this paper (Relation between two common localization methods for the EnKF, doi: 10.1007/s10596-010-9202-6), the author used a figure to

show the influence of B localization (covariance localization) method to the error co-variance matrix. The result of this localization method is mentioned in the paper. It said that "with non-zero elements centered around the matrix diagonal"(page 15, the first two lines). You only keep the elements in the matrix diagonal which means you only use the fixed localization scale. And also, in your "-local suffix " case, if I understand correctly, this is the "Local analysis" or "R localization" in data assimilation. When you design and set the influenced areas, you still used the fixed localization scale. Could you explain the reasons why you only use a fixed localization scale in your experiment set-up? Can you also explain how this localization scale was chosen? In the results part, the "local" case has a better performance compared with the "diagonal" case. Can the authors collaborate on the impact of different localization scales on the performance of DA? 14. In page 24, the last paragraph, the authors state that there are two ways to improve DA. A more realistic ensemble method to generate ensemble and observation correction algorithms can help to get better performance. These two conclusions are right. But, in your analysis part, you didn't compare the situation with specific ensemble generating method and the situation with generating ensemble randomly. In my opinion, no evidence in this paper can support this conclusion. Similarly, the second conclusion is not conclusive. Can you rephrase these two conclusions and make them open? In conclusion, after some modifications and additional explanations, I recommend accepting this paper.

---

## Referee Comment (RC2) · Anonymous Referee #2 · 16 Nov 2017

This study presents a data assimilation technique based on the Ensemble Kalman Filter, for reducing uncertainties in ISBA-CTRIP land surface model with ENVISAT radar altimeter based discharge data. The authors explored several localization methods during implementation in the Amazon basin, and made a good case, for the effectiveness of using Altimeter data to improve land surface models with the proper approach. I think the research presented here is in line with quality and value to its field that

Interactive
comment

HESS strives to provide for its readers. I do have a few concerns that I feel should be addressed before publication as well as a few minor suggestions. Major comments: 1. The objectives stated in line 5 and 6 of the abstract and those stated lines 23 -24 of page three are different. In my opinion, the paper demonstrates quite clearly that the assimilation improves results, but actually focuses in the difference between localization methods more than the importance of altimeter data as a source for reducing uncertainties. 2. The description of the altimetry based discharge product section (2.3.1) is quite in depth, however, it should really include a brief statement about QA/QC from the data source's literature. The instrument precision is provided, but the reader has no idea what sort of error that translates to in terms of discharge. 3. Page7 line 30-32 I'm curious what portion of virtual stations were associated with an adjacent cell. 4. Section 2.3.2, this draws further questions about the objective of the study. The authors point out that in situ data was not used in the assimilation. In my opinion, a comparative run with assimilated in situ data could help demonstrate the value of altimeter data, if that is the primary focus. 5. Page 24 lines 13-16, I think this should be clarified to be within topological limitation, (i.e. "should be impacted by all upriver observations"). 6. Page 24 line 19, This manuscript hasn't made a case to support the inclusion of discussion of the groundwater time constant as a major control on discharge. Please include information on this in the results section. 7. Page 24 lines 23-32, I think the authors need to be really careful assigning usefulness of these other altimeter mission for their assimilation protocol. The ENVISAT contemporary missions and those after, are likely to provide data quality that could allow for the construction of additional discharge data, but the casual mention of these missions doesn't really address the feasibility building rating curves and discharge data from them. The biggest issue here is the inclusion of earlier mission, and the citation provided. To my knowledge there has been only marginal success using pre-ENVISAT data on rivers. Using ERS 1-2 or TOPEX would most likely only work on the main channel if at all. In Tourian et al., 2017, the authors specifically mention that these earlier mission were not included because of poor inland performance. Minor comments: 1. Page 6 line 23, crosses the

river stream is redundant. 2. Page 23 line 2 "to correct directly the discharge" should be to directly correct the discharge. I think all of the clarifications and changes I have recommended are relatively minor. With a few changes, I will recommend accepting this paper for publication.

---

## Author Comment (AC1) · 31 Dec 2017

We would like to thank the reviewers for their very constructive comments. Please find below our replies.

*1. In the introduction, the third paragraph is about the resources of model errors. In my opinion, a description of the accumulation of model errors and other uncertainties in model's prediction which can lead to the collapse of the model should be added. I think this part can be regarded as a part of the explanation of the necessity of data assimilation. I recommend to shorten the*

[Figure]

*section on model error sources and add some sentences on the impact of accumulation of uncertainties.*

**Authors' reply:** Following your comment, we propose to reduce this paragraph by shortening the list of model uncertainty sources (which are the lack of knowledge on the real physics, the numerization/discretization induced errors and the uncertainties in the inputs parameters and forcings). Also, we will add an ending remark stating that the accumulation of all these uncertainties could lead to the model collapse if they become too important. Therefore, we propose to rephrase this part of the introduction as follow:

"However, even if hydrological models become more and more accurate, inherent model uncertainties are unavoidable. They originate from several sources: the lack of knowledge in the real physics, the numerization/discretization-induced errors and the uncertainties in the inputs parameters and forcings). All these uncertainties impact model's outputs. In the worst case, all those uncertainties could accumulate and result into the collapse of the model. The model gives therefore an approximate view of the system real state."

*2. In Lopez Lopez (2016) discharge data was assimilated into a model for the Rhine basin. I think this paper should be added to the introduction. (López López, P., Wanders, N., Schellekens, J., Renzullo, L. J., Sutanudjaja, E. H., and Bierkens, M. F. P.: Improved large-scale hydrological modelling through the assimilation of streamflow and downscaled satellite soil moisture observations, Hydrol. Earth Syst. Sci., 20, 3059-3076, https://doi.org/10.5194/hess-20-3059-2016, 2016)*

**Authors' reply:** Thank you for this suggestion. We will definitely add this citation in the

7th paragraph of the introduction, next to the citation of Trudel et al. (2014).

**3.** *Page 3, in last second paragraph, the aim of this study only mentioned that EnKF is applied and also said something about the model and observations. From my point of view, the using of localization methods should be mentioned. The localization methods are the crucial key to this case study and without localization, the academic value of this research will mostly be as a case study of an otherwise known method into a new geographical area.*

**Authors' reply:** We totally agree with the reviewer's suggestion. We are going to formulate the paper aims differently. The modified aims will state the following scientific question: how can we use remotely-sensed data at a river reach scale in order to improve a large-scale model. Because of the "local" information provided by the satellite and the model errors (and the difficulty to estimate them), the use of localization method is needed. Therefore, in the same paragraph, we will add some sentences to explicitly mention the need of "localization methods" in the quick method description. We propose to reformulate the objective as follows:
"The objective of the present study is to investigate the contribution of remote-sensed data that provide local information to improve a large-scale RRM via DA. The scale difference between the observations and the model lead to also study the need to use localization methods within our DA framework. We used an Ensemble Kalman Filter, to which we added a simple localization module, to assimilate discharges derived from ENVISAT water surface elevation measurements. These observations are used to correct the state of the large scale Total Runoff Integrated Pathways (TRIP, Oki and Sud, 1998) RRM version included in the land surface modelling platform "Surfaces Externalisées" (SurfEx, Masson et al., 2013), and developed at the Centre National de Recherches en Météorologie (CNRM, France). This particular version is denoted by the CTRIP acronym hereinafter. CTRIP is coupled with the Interactions-Soil-Biosphere-Atmosphere (ISBA, Boone et al., 1999) LSM at a resolution of $0.5° \times 0.5°$."

*4. In section 2, I think the description of the CTRIP RRM is excessive. It would be better to make the context of this model shorter and simpler. Maybe the authors can reference one of their earlier papers on the model and point to that for the model description.*

**Authors' reply:** As proposed by the reviewer, we will reduce the size of the CTRIP model description. First of all, we will remove the model equations along with the description of its parameters, as they are already described quite well in previous papers. Instead, the reader will be directed to these papers (especially: Decharme et al., 2010; Decharme et al., 2012). We will keep the description of the ISBA model as it is, as it is already quite synthetic.

*5. In section 3, it has the similar issue just like section 2, the explanation of the fairly standard EnKF is too long.*

**Authors' reply:** Similarly to the previous comment, we will reduce the length of the subsection by getting rid of the first paragraph that focuses on the classical Kalman Filter. So this subsection will directly start with the EnKF description and equations. We will also remove Eq.15, as the matrix in the equation is already described in the preceding text.

**6.** *Section 3.2.4 (Localization) does not belong to "3.2 Generating the ensembles". In my opinion, the following structure of section 3 is better. Firstly, introduce the control variables and observations separately. And then, give a short introduction to EnKF theory. Following, present how to implement the EnKF with localization specifically. Last part includes diagnostics and experiment set up.*

**Authors' reply:** Thank you for this suggestion, which will make the paper easier to read and highlight the localization method. We propose the following reorganisation of the third section:

- 3. Method

    - 3.1. Data assimilation variables
        - 3.1.1. Control variables
        - 3.1.2. Observations variables

    - 3.2. EnKF for ISBA-CTRIP state estimation
        - 3.2.1. EnKF theory
        - 3.2.2 Localization
        - 3.2.3. Generation of ensembles

    - 3.3. Assimilation diagnostics

**7.** *In page 14, section 3.2.4, the second paragraph, it said that there are three localization methods. I prefer to state that there are two common localization methods, namely local analysis (R localization) and covariance*

*localization (B localization) . These two methods can be found in following two papers. Balance and Ensemble Kalman Filter Localization Techniques (doi:10.1175/2010MWR3328.1) -Relation between two common localisation methods for the EnKF (doi: 10.1007/s10596-010-9202-6).*

**Authors' reply:** Thank you for the suggested publications. We will include them and present the two common localization methods in section 3.2.4.

We suggest the following changes to the section:

"It exists two types of localization techniques (Greybush et al., 2011; Sakov and Bertino, 2011). The first one is called B-localization. It is based on explicitly modifying the background error covariance matrix $\mathbf{P}^b_{e,k}$. The latter is multiplied by a correlation matrix generated from a radial function, namely a function of the two/three spatial dimensions which monotonously decreases with the distance between control variables (Hamill et al., 2001; Houtemaker and Mitchell, 2011; Houtemaker and Mitchell, 2005). A sparse matrix $\tilde{\mathbf{P}}^b_{e,k}$ is therefore computed, with non-zero elements centred around the matrix diagonal. This modified matrix replaces $\mathbf{P}^b_{e,k}$ in the calculation of the Kalman gain matrix $\mathbf{K}_{e,k}$. The other common localization technique is called R-localization or Local Analysis. This one consists in proceeding the analysis step into characteristic sub-spaces of the overall problem space."

*8.  The names of the different localization schemes should align with the common names in the data assimilation field. I recommend the terms in those two papers in point 7, above. In table 3 and corresponding parts in the main body of this research should change "-local suffix" to "Local analysis" or "R localization" and also replace "-diagonal suffix" with "covariance localization" or "B localization".*

**Authors' reply:** Given our understanding of Greybush et al. (2011) and Sakov and Bertino (2011) articles, it seems that we applied a B-localization method for both the experiment with suffix -diag and -local. The difference between these two experiments is the "localization matrix" multiplied (via the Schur product) to the background error covariance matrix. More explicitely, to apply the localization, we used the same equation than Moore (1973) and Biancamaria et al. (2011):

$$\mathbf{x}^a = \mathbf{x}^b + \left[\mathbf{S} \times \left(\mathbf{P}^b\mathbf{H}^T\right)\right]\left\{\mathbf{H}\left[\mathbf{S} \times \left(\mathbf{P}^b\mathbf{H}^T\right)\right] + \mathbf{R}\right\}^{-1}\left(\mathbf{y}^o - \mathbf{H}\mathbf{x}^b\right),$$

where $\mathbf{S}$ is the localization matrix and "$\times$" is the Schur product.
We will clearly state in the manuscript that we applied B-localization methods. As both methods are B-localization and we will explicitly mention it in the manuscript, we prefer not to change the names of the experiments.

*9. On page 16, the part before 3.3, it describes how to get the localization matrix. In this study, the author used localization matrix to multiply covariance matrix directly. This way is not wrong but it differs from the most common way to implement the localization methods. Can you use equations to display the formulation of localization matrix? This is helpful for readers to understand your localization methods.*

**Authors' reply:** We apologize for not giving the equation in the previous version of the manuscript. The equation we used (as mentionned in our previous comment) is taken from Moore (1973) and Biancamaria et al. (2011):

$$\mathbf{x}^a = \mathbf{x}^b + \left[\mathbf{S} \times \left(\mathbf{P}^b\mathbf{H}^T\right)\right]\left\{\mathbf{H}\left[\mathbf{S} \times \left(\mathbf{P}^b\mathbf{H}^T\right)\right] + \mathbf{R}\right\}^{-1}\left(\mathbf{y}^o - \mathbf{H}\mathbf{x}^b\right),$$

where **S** is the localization matrix and "×" is the Schur product. We will add this equation in the new section 3.2.2 ("Localization").

*10. In table 1, the size of the ensemble is 101. The authors do not justify the choice of exactly 101 ensemble members in the paper. It is not possible for the reviewer to see if the ensemble size represent the distribution of model states properly? Could the authors use some figures or the rank histogram of the ensemble to show the gaussianity of the ensemble? Otherwise, could the authors justify the choice for 101 ensemble members?*

**Authors' reply:** In our data assimilation platform, the ensemble size of exactly 101 instead of 100 is for implementation convenience. In the literature (e.g. Greybush et al., 2011) an ensemble size of 100 is often chosen to have a large ensemble, while maintaining a reasonable computational time of the data assimilation experiment. The suggestion of adding ensemble histogram is a very good suggestion, however, because the paper is already quite long, we prefer not to add such histograms in the present study. However, we will add this explanation in section 3.2.3 ("Generation of the ensembles") such as "To get a large ensemble, while maintaining a reasonable computational time, the ensemble size has been set to 101 members."

*11. In data assimilation applications with localization usually a common localization function is used. Common example is a fifth order function of Gaspari and Cohn. I didn't find the description of the localization function in this paper. If I missed it, please point out its location. If the authors didn't use it, could the authors explain reasons and considerations?*

**Authors' reply:** You are right, in the present study, we did not use such radial functions. The localization matrices we built only contain 0 or 1. The resulting "localized" error covariance matrices contain the exact same values as the error covariance matrices before localization, but only at elements corresponding to a "1" in the localization matrix. In a way, the localization only allows to suppress spurious effects of far away cells. The reason why we did not use functions such as the fifth order Gaspari function is that, in the literature, we only found use of this function in atmospheric and oceanographic applications where the study domain are full 2D/3D domain. It has also been used, to our knowledge, in river hydraulic model applied to a single reach (1D) river network with no modeled tributaries between the beginning and the outlet of the river reach (e.g. Biancamaria et al., 2011). What we want to highlight here is that the case of a complete river network is different from the atmospheric and oceanographic framework, as for this case discharge within the river network is not a 2D/3D field. It is a "multi-1D ramified network". Therefore, it appeared to us that we could not directly use the same localization method as in atmospheric and oceanographic applications (applied on a sphere or disk of influence). Using such kind of radial function is tricky, as they could only be applied locally over the different 1D river reaches and the case of pixels corresponding to connections with another tributary or with the mainstream is not trivial. That's why, we decided, as a first step to develop a localization method for a basin-scale river network application, to have a very simple localization function: the first developments of our localization method only select elements of the error covariance matrices but do not modify their value. Further development of the localization method may consider the use of such function within the river network. We discussed this difference between 2D/3D model atmospheric and "multi-1D" river model in the description of the localization method we used for our study (in section "Localization of the error covariance matrices").

**12.** *It is common that localization methods can cause imbalance. The analysis of imbalance can show the performance of localization method in specific application. I recommend adding the imbalance analysis. If the authors think it is unnecessary, could the authors explain the reasons?*

**Authors' reply:** Thank you for this suggestion. Greybush et al., (2011) clearly show the interest of an imbalance study. However, we do not think it is necessary for our present study. Indeed, the localization method do not use radial function (see our reply to the above comment), but just select elements in the error covariance matrices. Therefore, non-zero elements in the matrix are not modified. Moreover, the CTRIP model is a relatively simple model compared to complete atmospheric model used in Greybush et al., (2011). Indeed, CTRIP is based on a linear reservoir model. Its only prognostic variables are the water storage in 3 reservoirs and the discharge is deduced from the storage at each time step (diagnostic variable). Therefore, we think that the issues discussed in Greybush et al., (2011) should not be as critical in our study.
That's why we decided not to add the imbalance analysis. However, we suggest to raise the imbalance question in the paper perspectives.

**13.** *I am a bit confused about the chosen localization scales. The "diagonal error covariance matrix" in this paper is to apply B localization method to form the localized covariance matrix. In this paper (Relation between two common localization methods for the EnKF, doi: 10.1007/s10596-010-9202-6), the author used a figure to show the influence of B localization (covariance localization) method to the error covariance matrix. The result of this localization method is mentioned in the paper. It said that "with non-zero elements centered around the matrix diagonal"(page 15, the*

*first two lines). You only keep the elements in the matrix diagonal which means you only use the fixed localization scale. And also, in your "-local suffix " case, if I understand correctly, this is the "Local analysis" or "R localization" in data assimilation. When you design and set the influenced areas, you still used the fixed localization scale. Could you explain the reasons why you only use a fixed localization scale in your experiment set-up? Can you also explain how this localization scale was chosen? In the results part, the "local" case has a better performance compared with the "diagonal" case. Can the authors elaborate on the impact of different localization scales on the performance of DA?*

**Authors' reply:** What we wanted to do with the "-diag suffix" experiment is to test the behavior of the data assimilation platform in the extreme case of an error covariance matrix with a zero-correlation length (diagonal matrix). This experiment has the objective to illustrate the case when the model cells are completely independent from any other cells in the catchment (we only correct at the location of the observation). This experiment is then the opposite of the "-direct" experiment, where the correlation length can be considered as infinite (because there is no localization). From these two "extreme" experiments, we wanted to build an intermediary case: the "-local suffix" experiment. The local experiment can be seen as a "proof-of-concept" experiment, to show benefit of using observations to correct surrounding pixels. It should avoid the spurious correlations observed in the "-direct" experiment (due to the generation of the ensemble) and should improve results from "-diag" experiment, as observations correct more than one pixel. We chose a fixed localization scale for simplicity and as a first step in the feasibility study of the development of a localization method for a hydrology application.

The correlation lengths were determined with respect to the averaged flow velocity in the river. From the averaged flow velocity, we can deduce the traveled

distance over one assimilation window (which is one day here) and express it as a number of model grid cells. Therefore, for any given cell and over a day, the "area of influence" represent the set of upstream cells whose water flow will pass through the given cell during the day (which will therefore contribute to the discharge at the given cell) and the set of downstream cells that will receive the water from the given cell during the day (which will therefore receive discharge from this given cell).

Concerning the last question, to our understanding, both experiments (i.e. "-diag" and "-local") correspond to B-localization. In the local-experiment, the localization mask is more realistic as there are more than one cell impacted by the modification of a given cell. Therefore, the data assimilation results are logically better. We could add one or two sentences about this point in the discussion section. Finally, the main perspective of this study is to develop more elaborated correlations lengths (as it is discussed in the 4th paragraph of the paper conclusions). These more elaborated correlations lengths should be built by trying to localization method used in the literature, adapted to the specific case of a river network at a basin-scale (see our reply to comment number 11).

*14. In page 24, the last paragraph, the authors state that there are two ways to improve DA. A more realistic ensemble method to generate ensemble and observation correction algorithms can help to get better performance. These two conclusions are right. But, in your analysis part, you didn't compare the situation with specific ensemble generating method and the situation with generating ensemble randomly. In my opinion, no evidence in this paper can support this conclusion. Similarly, the second conclusion is not conclusive. Can you rephrase these two conclusions and make them open?*

**Authors' reply:** You are right, we should rephrase our last paragraph to make these two points more open. We will change the paragraph as follow:

"To improve these DA results, several aspects could be investigated. For example, it could be studied if a more realistic ensemble method generation could be helpful. In the present study, only the model initial condition and the precipitation forcing are perturbed to generate the background forecast ensemble. More uncertainties in this ensemble could be added by also perturbing CTRIP parameters and/or ISBA outputs. Another DA aspect to look into is the potential use of a smoothing data assimilation algorithm, such as the Ensemble Kalman Smoother (Evensen and Van leeuwen, 2000). A smoother could help to have less "variability" in the corrected discharge. Finally, the assimilation scheme presented in this study could be applied to other river basin in the world, as ISBA-CTRIP is a global LSM. However, more work is needed to apply the DA platform at global scale."

**References**

- Biancamaria, S., Durant, M., Andreadis, K. M., Bates, P. D., Boone, A., Mognard, N. M., Rodriguez, E., Alsdorf, D. E., Lettenmaier, D. P., and Clark, E. A. : Assimilation of virtual wide swath altimetry to improve Arctic river modeling, Remote Sensing of Environment, 115, 373–381, 2011.

- Boone, A., Calvet, J.-C., and Noilhan, J. : Inclusion of a Third Soil Layer in a Land Surface Scheme Using the Force-Restore Method, J. Hydrometeor., 38, 1611–1630, 1999.

- Evensen, G. and Leeuwen, P. J. V. : An Ensemble Kalman Smoother for Nonlinear Dynamics, Monthly Weather Review, 128, 1852–1867, 2000.

- Greybush, S. J., Kalnay, E., Miyoshi, T., Ide, K., and Hunt, B. R. : Balance

and Ensemble Kalman Filter Localization Techniques, Monthly Weather Review, 139, 511–522, 2011.

- Hamill, T. M., Whitaker, J. S., and Snyder, C. : Distance-Dependent Filtering of Background Error Covariance Estimates in an Ensemble Kalman Filter, Monthly Weather Review, 129, 2776–2790, 2001.

- Houtekamer, P. L. and Mitchell, H. L. : A sequential ensemble kalman filter for atmospheric data assimilation, Monthly Weather Review, 129, 124–137, 2001.

- Houtekamer, P. L. and Mitchell, H. L. : Ensemble Kalman filtering, Quarterly Journal of the Royal Meteorological Society, 131, 3269–3289, 2005.

- Masson, V., Moigne, P. L., Martin, E., Faroux, S., Alias, A., Alkama, R., Belamari, S., Barbu, A., Boone, A., Bouyssel, F., Brousseau, P., Brun, E., Calvet, J.-C., Carrer, D., Decharme, B., Delire, C., Donier, S., Essaouini, K., Gibelin, A.-L., Giordani, H., Habets, F., Jidane, M., Kerdraon, G., Kourzeneva, E., Lafayse, M., Lafont, S., Brossier, C. L., Lemonsu, A., Mafouf, J.-F., Marguinaud, P., Mokhtari, M., Morin, S., Pigeon, G., Salgado, R., Seity, Y., Taillefer, F., Tanguy, G., Tulet, P., Vincendon, B., Vionnet, V., and Voldoire, A. : The SURFEXv7.2 land and ocean surface platform for coupled or offline simulation of Earth surface variables and fluxes, Geoscientific Model Development, 6, 926–960, 2013.

- Moore, J. B. : Discrete-time fixed-lag smoothing algorithms, Automatica, 9, 163–173, 1973.

- Oki, T. and Sud, Y. C. : Design of Total Integrating Pathways (TRIP)-A Global River Channel Network, Earth Interactions, 2, 1–36, 1998.

- Sakov, P. and Bertino, L. : Relation between two common localisation methods for the EnKF, Computers and Geosciences, 15, 225–237, 2011.

---

## Author Comment (AC2) · 31 Dec 2017

Our replies to all the comments made by the reviewer can be found below. We thank the reviewer for all the comments and suggestions that helped to improve our manuscript.

**1 Major comments:**

*1.The objectives stated in line 5 and 6 of the abstract and those stated lines 23 -24 of page three are different. In my opinion, the paper demonstrates*

*quite clearly that the assimilation improves results, but actually focuses in the difference between localization methods more than the importance of altimeter data as a source for reducing uncertainties.*

**Authors' reply:** We completely understand the reviewer point of view and we agree that our objectives were not well described. What we want to do in this study is to show the contribution of nadir altimetry (punctual measurement) at the continental scale of a large catchment. Because of this context, the use of localization is required. Those two aspects (use of satellite-derived discharge and localization) are both equally important. Following the similar remark from the first referee, we will reformulate the paragraph in the introduction to highlight the importance of these two objectives. To homogenize the manuscript, we will also modify the abstract accordingly.

*2.The description of the altimetry based discharge product section (2.3.1) is quite in depth, however, it should really include a brief statement about QA/QC from the data source's literature. The instrument precision is provided, but the reader has no idea what sort of error that translates to in terms of discharge.*

**Authors' reply:** The quality assurance has been made by Paris et al., (2016) by constraining the rating curve coefficients within a physical range of values. They also conducted a sensitivity analysis that shows a small sensitivity of the coefficient estimation to first guess of the coefficient values. The quality check was done by comparing over a validation time period the satellite-derived discharge to the model discharge used to derive rating curve over a calibration period. Discharge was also compared to some in situ gages. Satellite-derived discharge is of course heavily correlated to the model accuracy. Overall, a comparison to 51 in situ measurements led to a mean Nash-Sutcliff coefficient (NS) around 0.8 and a Normalized Root Mean Square Error

(NRMSE) around 10% over the validation period (Table 8 in Paris et al., 2016). However, for upstream basins, results are not as good as for the main tributaries. Overall, when compared to MGB outputs and in situ time series), the mean NS is equal to 0.7 and the mean NRMSE to 10% (In the same paper Paris et al., (2016), a similar study has been led on the water elevations). This information have been added to the manuscript.

*3. Page7 line 30-32 I'm curious what portion of virtual stations were associated with an adjacent cell.*

**Authors' reply:** 19% (69 out of 367) of the VS have been associated to an adjacent cell. We will add this information in section 2.3.1 ("Altimetry-based discharge product").

*4. Section 2.3.2, this draws further questions about the objective of the study. The authors point out that in situ data was not used in the assimilation. In my opinion, a comparative run with assimilated in situ data could help demonstrate the value of altimeter data, if that is the primary focus.*

**Authors' reply:** We agree with the reviewer. But the objective of this study is not to show that the remote-sensed data is better for data assimilation than the in situ data. We want to show the contribution of the altimetry when used alone (with the objective to use it on ungaged catchment or with few up-to-date in situ gage time series). Then, in the present study, the in situ data are used as an alternative source of data to validate the assimilation results. Therefore, we have not added an experiment assimilating only in situ data.

**5.** *Page 24 lines 13-16, I think this should be clarified to be within topological limitation, (i.e. "should be impacted by all upriver observations").*

**Authors' reply:** Thank you for this suggestion. We will replace "should be impacted by all available observations" by "should be impacted by all upriver observations".

**6.** *Page 24 line 19, This manuscript hasn't made a case to support the inclusion of discussion of the groundwater time constant as a major control on discharge. Please include information on this in the results section.*

**Authors' reply:** This is true. However, to follow another remark from the first reviewer, this paragraph will be rewritten and will not mention the "groundwater time constant" anymore.

**7.** *Page 24 lines 23-32, I think the authors need to be really careful assigning usefulness of these other altimeter mission for their assimilation protocol. The ENVISAT contemporary missions and those after, are likely to provide data quality that could allow for the construction of additional discharge data, but the casual mention of these missions doesn't really address the feasibility building rating curves and discharge data from them. The biggest issue here is the inclusion of earlier mission, and the citation provided. To my knowledge there has been only marginal success using pre-ENVISAT data on rivers. Using ERS 1-2 or TOPEX would most likely only work on the main channel if at all. In Tourian et al., (2017), the authors specifically mention that these earlier mission were not included because of poor inland performance.*

**Authors' reply:** Thank you for this remark. We will delete in this paragraph references to pre-ENVISAT missions.

**2  Minor comments:**

*1. Page 6 line 23, crosses the river stream is redundant.*

**Authors' reply:** You are right. We will replace the expression by "crosses the river".

*2. Page 23 line 2 "to correct directly the discharge" should be to directly correct the discharge.*

**Authors' reply:** Thank you for noticing this mistake. We will replace "to correct directly the discharge" with "to directly correct the discharge".

**References**

- Paris, A., Paiva, R. C. D., Silva, J. S. D., Moreira, D., Calmant, S., Garambois, P.-A., Collischonn, W., Bonnet, M.-P., and Seyler, F. : Stage-discharge rating curves based on satellite altimetry and modeled discharge in the Amazon basin, Water Resources Research, 52, 2016.

---

## Author Response (AR1)

**Answer to reviewer Rolf Hut and Zhenwu Wang**

C. M. Emery, A. Paris, S. Biancamaria, A. Boone, S. Calmant, P.-A. Garambois, J. S. D. Silva

24 janvier 2018

We would like to thank the reviewers for their very constructive comments. Please find below our replies.

Besides, the manuscript changes linked to those comments will be in orange in the manuscript.

1. In the introduction, the third paragraph is about the resources of model errors. In my opinion, a description of the accumulation of model errors and other uncertainties in model's prediction which can lead to the collapse of the model should be added. I think this part can be regarded as a part of the explanation of the necessity of data assimilation. I recommend to shorten the section on model error sources and add some sentences on the impact of accumulation of uncertainties.

**Authors' reply :** Following your comment, we propose to reduce this paragraph by shortening the list of model uncertainty sources (which are the lack of knowledge on the real physics, the numerization/-discretization induced errors and the uncertainties in the inputs parameters and forcings). Also, we will add an ending remark stating that the accumulation of all these uncertainties could lead to the model collapse if they become too important. Therefore, we propose to rephrase this port of the introduction as follow :

"However, even if hydrological models become more and more accurate, inherent model uncertainties are unavoidable. They originate from several sources : the lack of knowledge in the real physics, the numerization/discretization-induced errors and the uncertainties in the inputs parameters and forcings). All these uncertainties impact model's outputs. In the worst case, all those uncertainties could accumulate and result into the collapse of the model. The model gives therefore an approximate view of the system real state."

**2.** In Lopez Lopez (2016) discharge data was assimilated into a model for the Rhine basin. I think this paper should be added to the introduction. (López López, P., Wanders, N., Schellekens, J., Renzullo, L. J., Sutanudjaja, E. H., and Bierkens, M. F. P. : Improved large-scale hydrological modelling through the assimilation of streamflow and downscaled satellite soil moisture observations, Hydrol. Earth Syst. Sci., 20, 3059-3076, https://doi.org/10.5194/hess-20-3059-2016, 2016)

**Authors' reply :**Thank you for this suggestion. We will definitely add this citation in the 7th paragraph of the introduction, next to the citation of Trudel et al. (2014).

**3.** Page 3, in last second paragraph, the aim of this study only mentioned that EnKF is applied and also said something about the model and observations. From my point of view, the using of localization methods should be mentioned. The localization methods are the crucial key to this case study and without localization, the academic value of this research will mostly be as a case study of an otherwise known method into a new geographical area.

**Authors' reply** :We totally agree with the reviewer's suggestion. We are going to formulate the paper aims differently. The modified aims will state the following scientific question : how can we use remotelysensed data at a river reach scale in order to improve a large-scale model. Because of the "local" information provided by the satellite and the model errors (and the difficulty to estimate them), the use of localization method is needed. Therefore, in the same paragraph, we will add some sentences to explicitly mention the need of "localization methods" in the quick method description. We propose to reformulate the objective as follows :

"The objective of the present study is to investigate the contribution of remote-sensed data that provide local information to improve a large-scale RRM via DA. The scale difference between the observations and the model lead to also study the need to use localization methods within our DA framework. We used an Ensemble Kalman Filter, to which we added a simple localization module, to assimilate discharges derived from ENVISAT water surface elevation measurements. These observations are used to correct the state of the large scale Total Runoff Integrated Pathways (TRIP, Oki and Sud, 1998) RRM version included in the land surface modelling platform "Surfaces Externalisées" (SurfEx, Masson et al., 2013), and developed at the Centre National de Recherches en Météorologie (CNRM, France). This particular version is denoted by the CTRIP acronym hereinafter. CTRIP is coupled with the Interactions-Soil-Biosphere-Atmosphere (ISBA, Boone et al., 1999) LSM at a resolution of  $0.5^{\circ} \times 0.5^{\circ}$ ."

**4.** In section 2, I think the description of the CTRIP RRM is excessive. It would be better to make the context of this model shorter and simpler. Maybe the authors can reference one of their earlier papers on the model and point to that for the model description.**

**Authors' reply :**As proposed by the reviewer, we will reduce the size of the CTRIP model description. First of all, we will remove the model equations along with the description of its parameters, as they are already described quite well in previous papers. Instead, the reader will be directed to these papers (especially : Decharme et al., 2010; Decharme et al., 2012). We will keep the description of the ISBA model as it is, as it is already quite synthetic.

**5.** In section 3, it has the similar issue just like section 2, the explanation of the fairly standard EnKF is too long.**

**Authors' reply :**Similarly to the previous comment, we will reduce the length of the subsection by getting rid of the first paragraph that focuses on the classical Kalman Filter. So this subsection will directly start with the EnKF description and equations. We will also remove Eq.15, as the matrix in the equation is already described in the preceding text.

**6.** Section 3.2.4 (Localization) does not belong to "3.2 Generating the ensembles". In my opinion, the following structure of section 3 is better. Firstly, introduce the control variables and observations separately. And then, give a short introduction to EnKF theory. Following, present how to implement the EnKF with localization specifically. Last part includes diagnostics and experiment set up.

**Authors' reply** :Thank you for this suggestion, which will make the paper easier to read and highlight the localization method. We propose the following reorganisation of the third section :

- 3. Method
  - 3.1. Data assimilation variables
    - 3.1.1. Control variables
    - 3.1.2. Observations variables
  - 3.2. EnKF for ISBA-CTRIP state estimation
    - 3.2.1. EnKF theory
    - 3.2.2 Localization
    - 3.2.3. Generation of ensembles
  - 3.3. Assimilation diagnostics

**7.** In page 14, section 3.2.4, the second paragraph, it said that there are three localization methods. I prefer to state that there are two common localization methods, namely local analysis (R localization) and covariance localization (B localization). These two methods can be found in following two papers. Balance and Ensemble Kalman Filter Localization Techniques (doi :10.1175/2010MWR3328.1) -Relation between two common localisation methods for the EnKF (doi : 10.1007/s10596-010-9202-6).

**Authors' reply :**Thank you for the suggested publications. We will include them and present the two common localization methods in section 3.2.4.

We suggest the following changes to the section :

"It exists two type of localization techniques (Greybush et al., 2011; Sakov and Bertino, 2011). The first one is called B-localization. It is based on explicitly modifying the background error covariance matrix  $\mathbf{P}_{e,k}^{b}$ . It consists in multiplying the matrix  $\mathbf{P}_{e,k}^{b}$  by a correlation matrix generated from a radial function, namely a function of the two/three spatial dimensions which monotonously decreases with the distance between control variables (Hamill et al., 2001; Houtekamer and Mitchell, 2001, 2005). A sparse matrix  $\tilde{\mathbf{P}}_{e,k}^{b}$  is therefore computed, with non-zero elements centred around the matrix diagonal. This modified matrix replaces  $\mathbf{P}_{e,k}^{b}$  in the calculation of the Kalman gain matrix  $\mathbf{K}_{e,k}$ . The other common localization technique is called R-localization or local analysis. This one consists in proceeding the analysis step into characteristic sub-spaces of the overall problem space."

**8.** The names of the different localization schemes should align with the common names in the data assimilation field. I recommend the terms in those two papers in point 7, above. In table 3 and corresponding parts in the main body of this research should change "-local suffix" to "Local analysis" or "R localization" and also replace "-diagonal suffix" with "covariance localization" or "B localization".

Authors' reply : Given our understanding of Greybush et al. (2011) and Sakov and Bertino (2011) articles, it seems that we applied a B-localization method for both the experiment with suffix -diag and

-local. The difference between these two experiment is the "localization matrix" multiplied (via the Schur product) to the background error covariance matrix. More explicitly, to apply the localization, we used the same equation than Moore (1973); Biancamaria et al. (2011) :

$$\mathbf{x}^{a} = \mathbf{x}^{b} + \left[\mathbf{S} imes \left(\mathbf{P}^{b} \mathbf{H}^{\mathsf{T}}
ight)
ight] \left\{\mathbf{H} \left[\mathbf{S} imes \left(\mathbf{P}^{b} \mathbf{H}^{\mathsf{T}}
ight)
ight] + \mathbf{R}
ight\}^{-1} \left(\mathbf{y}^{o} - \mathbf{H} \mathbf{x}^{b}
ight),$$

where  $\mathbf{S}$  is the localization matrix and " $\times$ " is the Schur product.

We will clearly state in the manuscript that we applied B-localization methods. As both methods are B-localization and we will explicitly mention it in the manuscript, we prefer not to change the names of the experiments.

**9.** On page 16, the part before 3.3, it describes how to get the localization matrix. In this study, the author used localization matrix to multiply covariance matrix directly. This way is not wrong but it differs from the most common way to implement the localization methods. Can you use equations to display the formulation of localization matrix? This is helpful for readers to understand your localization methods.

**Authors' reply**: We apologize for not giving the equation in the previous version of the manuscript. The equation we used (as mentionned in our previous comment) is taken from Moore (1973); Biancamaria et al. (2011) :

$$\mathbf{x}^{a} = \mathbf{x}^{b} + \left[\mathbf{S} imes \left(\mathbf{P}^{b} \mathbf{H}^{\mathsf{T}}
ight)
ight] \left\{\mathbf{H} \left[\mathbf{S} imes \left(\mathbf{P}^{b} \mathbf{H}^{\mathsf{T}}
ight)
ight] + \mathbf{R}
ight\}^{-1} \left(\mathbf{y}^{o} - \mathbf{H} \mathbf{x}^{b}
ight),$$

where S is the localization matrix and " $\times$ " is the Schur product. We will add this equation in the new section 3.2.2 ("Localization").

**10.** In table 1, the size of the ensemble is 101. The authors do not justify the choice of exactly 101 ensemble members in the paper. It is not possible for the reviewer to see if the ensemble size represent the distribution of model states properly? Could the authors use some figures or the rank histogram of the ensemble to show the gaussianity of the ensemble? Otherwise, could the authors justify the choice for 101 ensemble members?

**Authors' reply** :In our data assimilation platform, the ensemble size of exactly 101 instead of 100 is for implementation convenience. In the litterature (e.g. Greybush et al. (2011)) an ensemble size of 100 is often chosen to have a large ensemble, while maintaining a reasonable computational time of the data assimilation experiment. The suggestion of adding ensemble histogram is a very good suggestion, however, because the paper is already quite long, we prefer not to add such histograms in the present study. However, we will add this explanation in section 3.2.3 ("Generation of the ensembles") : "To get a large ensemble, while maintaining a reasonable computational time, the ensemble size has been set to 101 members."

**11.** In data assimilation applications with localization usually a common localization function is used. Common example is a fifth order function of Gaspari and Cohn. I didn't find the description of the localization function in this paper. If I missed it, please point out its location. If the authors didn't use it, could the authors explain reasons and considerations?

Authors' reply : You are right, in the present study, we did not use such radial functions. The localization matrices we built only contain 0 or 1. The resulting "localized" error covariance matrices contain the exact same values as the error covariance matrices before localization, but only at elements corresponding to a "1" in the localization matrix. In a way, the localization only allows to suppress spurious effects of far away cells. The reason why we did not use functions such as the fifth order Gaspari function is that, in the litterature, we only found use of this function in atmospheric and oceanographic applications where the study domain are full 2D/3D domain. It has also been used, to our knowledge, in river hydraulic model applied to a single reach (1D) river network with no modeled tributaries between the beginning and the outlet of the river reah (e.g. Biancamaria et al. (2011)). What we want to highlight here is that the case of a complete river network is different from the atmospheric and oceanographic framework, as for this case discharge within the river network is not a 2D/3D field. It is a "multi-1D ramified network". Therefore, it appeared to us that we could not directly use the same localization method as in atmospheric and oceanographic applications (applied on a sphere or disk of influence). Using such kind of radial function is trikky, as they could only be applied locally over the different 1D river reaches and the case of pixels corresponding to connections with another tributary or with the mainstream is not trivial. That's why, we decided, as a first step to develop a localization method for a basin-scale river network application, to have a very simple localization function : the first developments of our localization method only select elements of the error covariance matrices but do not modify their value. Further development of the localization method may consider the use of such function within the river network. We discussed this difference between 2D/3D model atmospheric and "multi-1D" river model in the description of the localization method we used for our study (in section "Localization of the error covariance matrices").

**12.** It is common that localization methods can cause imbalance. The analysis of imbalance can show the performance of localization method in specific application. I recommend adding the imbalance analysis. If the authors think it is unnecessary, could the authors explain the reasons?**

**Authors' reply :**Thank you for this suggestion. Greybush et al. (2011) clearly show the interest of an imbalance study. However, we do not think it is necessary for our present study. Indeed, the localization method do not use radial function (see our reply to the above comment), but just select elements in the error covariance matrices. Therefore, non-zero elements in the matrix are not modified. Moreover, the CTRIP model is a relatively simple model compared to complete atmospheric model used in Greybush et al. (2011). Indeed, CTRIP is based on a linear reservoir model. Its only prognostic variables are the water storage in 3 reservoirs and the discharge is deduced from the storage at each time step (diagnostic variable). Therefore, we think that the issues discussed in Greybush et al. (2011) should not be as critical in our study.

That's why we decided not to add the imbalance analysis. However, we suggest to raise the imbalance question in the paper perspectives.

**13.** I am a bit confused about the chosen localization scales. The "diagonal error covariance matrix" in this paper is to apply B localization method to form the localized covariance matrix. In this paper (Relation between two common localization methods for the EnKF, doi : 10.1007/s10596-010-9202-6), the author used a figure to show the influence of B localization

(covariance localization) method to the error covariance matrix. The result of this localization method is mentioned in the paper. It said that "with non-zero elements centered around the matrix diagonal"(page 15, the first two lines). You only keep the elements in the matrix diagonal which means you only use the fixed localization scale. And also, in your "-local suffix " case, if I understand correctly, this is the "Local analysis" or "R localization" in data assimilation. When you design and set the influenced areas, you still used the fixed localization scale. Could you explain the reasons why you only use a fixed localization scale in your experiment set-up? Can you also explain how this localization scale was chosen? In the results part, the "local" case has a better performance compared with the "diagonal" case. Can the authors elaborate on the impact of different localization scales on the performance of DA?

**Authors' reply** :What we wanted to do with the "-diag suffix" experiment is to test the behavior of the data assimilation platform in the extreme case of an error covariance matrix with a zero-correlation length (diagonal matrix). This experiment has the objective to illustrate the case when the model cells are completely independant from any other cells in the catchment (we only correct at the location of the observation). This experiment is then the opposite of the "-direct" experiment, where the correlation length can be considered as infinite (because there is no localization). From these two "extreme" experiments, we wanted to build an intermediary case : the "-local suffix" experiment. The local experiment can be seen as a "proof-of-concept" experiment, to show benefit of using observations to correct surrounding pixels. It should avoid the spurious correlations observed in the "-direct" experiment (due to the generation of the ensemble) and should improve results from "-diag" experiment, as observations correct more than one pixel. We chose a fixed localization scale for simplicity and as a first step in the feasibility study of the development of a localization method for a hydrology application.

The correlation lengths were determined with respect to the averaged flow velocity in the river. From the averaged flow velocity, we can deduce the travelled distance over one assimilation window (which is one day here) and express it as a number of model grid cells. Therefore, for any given cell and over a day, the "area of influence" represent the set of upstream cells whose water flow will pass through the given cell during the day (which will therefore contribute to the discharge at the given cell) and the set of downstream cells that will receive the water from the given cell during the day (which will therefore receive discharge from this given cell).

Concerning the last question, to our understanding, both experiments (i.e. "-diag" and "-local") correspond to B-localization. In the local-experiment, the localization mask is more realistic as there are more than one cell impacted by the correction from an observation. Therefore, the data assimilation results are logically better. We could add one or two sentences about this point in the discussion section. Finally, the main perspective of this study is to develop more elaborated correlations lengths (as it is discussed in the 4th paragraph of the paper conclusions). These more elaborated correlations lengths should be built by trying to localization method used in the literature, adapted to the specific case of a river network at a basin-scale (see our reply to comment number 11).

14. In page 24, the last paragraph, the authors state that there are two ways to improve DA. A more realistic ensemble method to generate ensemble and observation correction algorithms can help to get better performance. These two conclusions are right. But, in your analysis part, you didn't compare the situation with specific ensemble generating method and the situation with generating ensemble randomly. In my opinion, no evidence in this paper can support this conclusion. Similarly, the second conclusion is not conclusive. Can you rephrase these two conclusions and make them open?

**Authors' reply :**You are right, we should rephrase our last paragraph to make these two points more open. We will change the paragraphe as follow :

"To improve these DA results, several aspects could be investigated. For example, it could be studied if a more realistic ensemble method generation could be helpful. In the present study, only the model initial condition and the precipitation forcing are perturbed to generate the background forecast ensemble. More uncertainties in this ensemble could be added by also perturbing CTRIP parameters and/or ISBA outputs. Another DA aspect to look into is the potential use of a smoothing data assimilation algorithm, such as the Ensemble Kalman Smoother (Evensen and Leeuwen, 2000). A smoother could help to have less "variability" in the corrected discharge. Finally, the assimilation scheme presented in this study could be applied to other river basin in the world, as ISBA-CTRIP is a global LSM. However, more work is needed to apply the DA platform at global scale."

[revised manuscript text omitted]

We obtain that (for more details on this approximation, see appendix A):

$$S_{\text{end},k,i,[\text{kg}]}^{a} \approx \rho L W^{2/5} s^{-3/10} n^{3/5} \left( Q_{k,i,[\text{kg.s}^{-1}]}^{a} \right)^{3/5},\tag{5}$$

with  $\rho$  [m3.kg-1] the water density, L [m] the river section length, W [m] the river width, s [-] the riverbed slope and n [-] the Manning coefficient in the riverbed. Then, for experiments with discharges as control variables, the formula in Eq. 3.1.1 will be used to convert corrected discharges into river stock and then propagate the model to the next observation time.

**- Option 2: correcting ISBA-CTRIP final water stock**

For this option, the control variables, gathered into the vector  $\mathbf{x}_k$  are the ISBA-CTRIP final water stock  $S_{\text{end},i,k}$ ,  $i = 1 \dots n_x$  estimated for each 2028 cells in the TRIP Amazon basin, at the assimilation cycle k.

The computational cost for this option is the same as for the first option but, now, the observation operator is defined as

$$\qquad \mathcal{H}_k = \mathcal{S}_k \circ \mathcal{Z}_k, \tag{6}$$

where  $\mathcal{Z}_k$ , is the operator (implicitly defined within TRIP) that turns the river final stock  $S_{\text{end},i,k}$  into equivalent discharge  $Q_{i,k}$ . Even though  $\mathcal{H}_k$  is not linear any more, this option presents the advantage of correcting the river final stock  $S_{k,\text{end}}^a$  that can be directly used for the next assimilation cycle and no additional uncertainties are introduced. However, the corresponding analysis discharge  $Q_{i,k}^a$  is now unknown as the explicit expression of  $\mathcal{Z}_k$  is also unknown. A potential formula to determine  $Q_{i,k}^a$  can be deduced from Eq. 3.1.1. Such a formula will be necessary to make comparative statistics to ENVISAT and in situ discharges and be able to evaluate the assimilation performances.

**- Option 3: correcting ISBA-CTRIP initial water stock**

For this final option, the control variables, gathered into the vector  $\mathbf{x}_k$  are the ISBA-CTRIP initial water stock  $S_{\text{ini},i,k}$ ,  $i = 1 \dots n_x$  estimated for each 2028 cells in the TRIP Amazon basin, at the assimilation cycle k.

The discharge observations are used to correct the surface water stock at the time prior to the observation time or, in other words, at the initial time of the integrating window. Therefore, the observation operator is written as the composition of the model operator  $\mathcal{M}_{[k-1,k]}$  with the observation operator defined in Eq. 6:

$$\mathcal{H}_k = \mathcal{S}_k \circ \mathcal{Z}_k \circ \mathcal{M}_{[k-1,k]}. \tag{7}$$

5

This operator is highly non-linear as it contains the full model operator. However, it is the only option where no uncertainties are added from the use of an external formula to compute corrected discharge at the observation time. Uncertainties in the stock-discharge relationship are only due to the model uncertainties. Indeed, once the analysis initial water stock  $S^a_{\text{ini},k,i}$  is determined, the control variables update must be propagated through the next assimilation time by re-integrating the ISBA-CTRIP model over the assimilation window. The initial water stock  $S^a_{\text{end},k,i}$  and the analysis discharge  $Q^a_{k,i}$ are automatically determined during this run.

15

25

30

**3.1.2 Observation variables**

In the framework of the state estimation, the observation variables, at a given day within the Amazon basin, are the discharge estimates derived from ENVISAT water surface elevations at the virtual stations associated to an ISBA-CTRIP cell. The ENVISAT repeatability is 35 days, therefore a given virtual station will provide an observation every 35 days at best. During the data assimilation experiments, all virtual stations will be used simultaneously. Because of the ENVISAT orbit, the number

of available observations at a given day will vary between 0 and 15, and these observations will be assimilated daily via the EnKF. Then, the observation vector  $\mathbf{y}_k^o$  at the assimilation cycle k (equivalently, at the simulation day k) is composed of the  $n_{y,k}$  discharge measures available at day k:

$$\mathbf{y}_{k}^{o} = \begin{bmatrix} y_{k,1}^{o}, & y_{k,2}^{o}, & \dots, & y_{k,n_{y,k}}^{o}, \end{bmatrix}$$
(8)

20 where  $y_{k,j}^{o}$ ,  $j = 1 \dots n_{y,k}$  is the *j*-th observation among the  $n_{y,k}$  at cycle k.

Measurement errors  $\epsilon_{\mathbf{k}}^{\mathbf{m}}$  come from errors in ENVISAT water surface elevations, errors in MGB discharges and uncertainties in the rating curves parameters used to turn water surface elevation into discharge. Sorooshian and Dracup (1980), Clark et al. (2008) and Paris et al. (2016) noticed that the concavity of the elevation-discharge relationship implies that the higher a water elevation, is the more uncertain the corresponding discharge. Therefore, the observation error standard deviation  $\sigma_{k,j}^o$ , associated to the *j*-th observation at cycle *k*, is defined with respect to the instantaneous discharge measure  $y_{k,j}^o$ , i.e.:

$$\sigma_{k,j}^o = \eta_j^o \times y_{k,j}^o, \ j = 1 \dots n_{y,k} \tag{9}$$

where  $\eta_j^o \in ]0,1[$  is a constant depending on the virtual station, such that  $\eta_j^o$  models the relative error. The observation error standard deviation  $\sigma_{k,j}^o$  is then a fraction of the instantaneous discharge. For each virtual station, the value of  $\eta_j^o$  depends on, first, the deviation from the MGB discharge, noted  $\sigma_j^{\text{mgb}}$  [%] and determined by Paris et al. (2016). As MGB discharges were used to determine ENVISAT discharges from ENVISAT water elevations,  $\sigma_j^{\text{mgb}}$  represents the deviation between the two

discharges data. Second, to take into account that MGB discharge is not perfect (in other words, to take into account some deviation from the real discharge), 0.05 is added to  $\sigma_i^{\text{mgb}}$  and the sum is rounding up to the nearest whole number, giving

$$\eta_j^{\rm mgb} = {\rm E}\left(\sigma_j^{\rm mgb} + 0.05\right), \label{eq:eq:eq:mgb}$$

[revised manuscript text omitted]
}^b_{e,k}) - \overline{\mathcal{H}(\mathbf{X}^b_{\bullet,k})} \cdot \mathbb{1}^T_{n_e} \right) \left( \mathcal{H}(\mathbf{X}^b_{e,k}) - \overline{\mathcal{H}(\mathbf{X}^b_{\bullet,k})} \cdot \mathbb{1}^T_{n_e} \right)^T,$$
(B2)

with  $\mathbf{X}^{b}_{e,k}$  the control matrix containing the  $n_e$  control vector  $\mathbf{x}^{b,[l]}_k$ ,  $l = 1 \dots n_e$  from the background ensemble such that

10
$$\mathbf{X}_{e,k}^{b} = \begin{bmatrix} \mathbf{x}_{k}^{b,[1]} \ \dots \ \mathbf{x}_{k}^{b,[N_{e}]} \end{bmatrix}$$

and  $\mathcal{H}(\mathbf{X}^b_{e,k})$  the same control matrix expressed in the observation space such that

$$\mathcal{H}(\mathbf{X}_{e,k}^{b}) = \left[\mathcal{H}(\mathbf{x}_{k}^{b,[1]}) \ \dots \ \mathcal{H}(\mathbf{x}_{k}^{b,[n_{e}]})\right].$$

 $\overline{\mathbf{X}_{\bullet,k}^{b}}$  and  $\overline{\mathcal{H}(\mathbf{X}_{\bullet,k}^{b})}$  are the ensemble sample expectations of the control matrix  $\mathbf{X}_{e,k}^{b}$  and its mapping on the observation space  $\mathcal{H}(\mathbf{X}_{e,k}^{b})$  respectively such that

15
$$\overline{\mathbf{X}^{b}_{\bullet,k}} = \frac{1}{n_e} \sum_{l=1}^{n_e} \mathbf{x}^{b,[l]}_k \quad \overline{\mathcal{H}(\mathbf{X}^{b}_{\bullet,k})} = \frac{1}{n_e} \sum_{l=1}^{n_e} \mathcal{H}(\mathbf{x}^{b,[l]}_k).$$

The vectors dimension are  $n_x$  and  $n_{y,k}$  respectively and  $\mathbb{1}_{n_e}$  is a vector of size  $n_e$  containing only 1s.

**Appendix C: Perturbations of precipitations**

The ensemble of perturbed precipitation fields  $\widetilde{\mathbf{F}}_e$  is defined such that:

$$\widetilde{\mathbf{F}}_{e} = \left\{ \widetilde{\mathbf{F}}^{[1]}, \quad \widetilde{\mathbf{F}}^{[2]}, \quad \dots \quad \widetilde{\mathbf{F}}^{[n_{e}]} \right\} = \left\{ \varphi_{p}^{[1]} \cdot \mathbf{F}, \quad \varphi_{p}^{[2]} \cdot \mathbf{F}, \quad \dots \quad \varphi_{p}^{[n_{e}]} \cdot \mathbf{F} \right\}, \tag{C1}$$

20 where:

[revised manuscript text omitted]